# An electrochemiluminescence device powered by streaming potential for the detection of amines in flowing solution

Rintaro Suzuki[1,6], Suguru Iwai [ID][1,6], Ryota Kirino[1,6], Kosuke Sato[1], Mariko Konishi[1], George Hasegawa [ID][2], Norio Ishizuka[3], Kimihiro Matsukawa[4], Kazuo Tanaka [ID][5], Elena Villani [ID][1] [✉] & Shinsuke Inagi [ID][1] [✉]

The research and implementation of portable and low-cost analytical devices that possess high reproducibility and ease of operation is still a challenging task, and a growing field of importance, within the analytical research. Herein, we report the concept, design and optimization of a microfluidic device based on electrochemiluminescence (ECL) detection that can be potentially operated without electricity for analytical purposes. The device functions exploiting the concept of streaming potential-driven bipolar electrochemistry, where a potential difference, generated from the flow of an electrolyte through a microchannel under the influence of a pressure gradient, is the driving force for redox reactions. To our purpose, we employ such a device to drive the ECL reaction of an organic chromophore deposited onto the electrode surface by simply flowing an electrolytic solution containing a coreactant into the microfluidic system, and we successively apply such device for the detection of amines in water. Our device shows high reproducibility and satisfactory detection limits for tri-$n$-propylamine, demonstrating an original, and up to now unexplored, concept of energy saving microfluidic systems with integrated ECL detection.

Electrochemiluminescence (ECL) continues to grow in importance as a highly sensitive and versatile electroanalytical technique in both academic research[1,2] and the healthcare diagnostic market[3,4]. This technology is based on a light generation that occurs at the electrode surface after an exergonic electron-transfer reaction between a chromophore and a sacrificial species called "coreactant", that is, a reactant that upon oxidation or reduction generates radicals sufficiently stable to react with the reduced or oxidized form, respectively, of the chromophore[5–7]. Because an electrochemical stimulus triggers the luminescence, no external light source is required to generate the chromophore's emitting excited state, implying that the ECL

technology has virtually no background signal and, hence, high sensitivity. Moreover, the precise spatial and temporal control over the light emission reaction enables the remote imaging of biological entities and the mapping of the electrochemical reactivity at the electrode surface[8–10]. All these intrinsic characteristics make ECL a highly popular and easy-to-use detection tool for a wide range of analytes[11–14].

It should also be important to underline that part of the great success of ECL is also due to the simple and relatively inexpensive equipment necessary to perform an ECL assay[2]. Generally, the basic setup comprises a potentiostat, an electrochemical cell, a light

[1]Department of Chemical Science and Engineering, School of Materials and Chemical Technology, Institute of Science Tokyo, Nagatsuta-cho, Midori-ku, Yokohama, Japan. [2]Institute of Materials and Systems for Sustainability, Nagoya University, Furo-cho, Chikusa-ku, Nagoya, Japan. [3]Emaus Kyoto Inc., Nishida-cho, Saiin, Ukyo-ku, Kyoto, Japan. [4]Materials Innovation Lab, Kyoto Institute of Technology, Goshokaido-cho, Matsugasaki, Sakyo-ku, Kyoto, Japan. [5]Department of Polymer Chemistry, Graduate School of Engineering, Kyoto University, Katsura, Nishikyo-ku, Kyoto, Japan. [6]These authors contributed equally: Rintaro Suzuki, Suguru Iwai, Ryota Kirino. [✉]e-mail: villani.e@mct.isct.ac.jp; inagi@cap.mac.titech.ac.jp

collector, e.g., a photomultiplier tube (PMT) or a digital camera, and a dark box, i.e., a light-tight compartment to screen the light collector from external photons[2,15,16]. In this regard, the few hardware requirements of ECL are of great advantage for its integration into miniaturized systems specifically designed for sensing purposes, with the aim of achieving low limits of detection while reducing sample volumes. Indeed, some examples of such integrated systems have been recently reported. For instance, smartphones with built-in camera technology having excellent sensing capabilities for analytical purposes can be used for ECL detection, providing real-time analysis through the fusion of widespread consumer hardware with networked technologies[17]. Similarly, microfluidic paper-based ECL devices combine minimal reagent consumption with a cheap and disposable platform for fast detection in a single portable instrument[18]. In particular, the integration of ECL detection into microfluidic systems has been reported as an effective method for carrying out highly sensitive analysis in low sample volume, and by employing portable devices with high reproducibility and ease of operation[19]. Therefore, the continuous search for innovative approaches to combine microfluidics with ECL detection is still a challenging goal to address towards the implementation of analytical strategies aiming at cost reduction and minimal human handling.

Herein, we report the design, development and optimization of an ECL device powered by streaming potential for the detection of amines in flowing solution that can be activated without the need of electric power or a potentiostat (Fig. 1). The device operates exploiting the concept of streaming potential-driven bipolar electrochemistry[20], that is a laminar flow of electrolytic solution passing through a microchannel with charged walls under a pressure gradient causes a charge imbalance from which a streaming potential ($E_{str}$) is originated[21]. Because the magnitude of such $E_{str}$ can reach up to 2–3 V[22], it can act as a driving force for faradaic reactions, namely, the ECL reaction can be achieved without the use of a potentiostat or electricity. In our study, we have investigated several experimental parameters to find the

optimal operating conditions to drive the ECL reaction, including the type of filling material, different types of solvent compositions and different flow rates. Finally, we have tested our device for the ECL detection of amines, because they are key compounds that behave as coreactants for light generation[23–25] and also constitute an important class of environmental pollutants that bear toxic potential for human exposure[26]. However, this concept can be extended for the ECL detection of a larger pool of analytes, including those for environmental monitoring, food and water testing and biowarfare agents. Furthermore, our prototype allows doing electrochemistry without the use of the most obvious equipment, that is a potentiostat, with a close look to cost savings.

## Results

### Setup for the streaming potential ECL device
Our custom-made device is a microfluidic system composed of two distinct chambers made of polyether ether ketone (PEEK) communicating through a PEEK tube containing a filler. Each chamber accommodates a platinum (Pt) wire, and the two are connected to each other with an ammeter (A in Fig. 1) outside the fluidic space, realizing a split bipolar electrode (split BPE) system. According to the principles of bipolar electrochemistry[27,28], a split BPE system behaves as a continuous BPE[29–31], implying that one Pt wire behaves as an anode, whereas the other behaves as a cathode. Fresh electrolytic solution is injected through the inlet channel in one of the cell compartment by using a plunger pump and, throughout the tube, flows directly to the second cell compartment before being collected from the outlet channel as a waste solution (Fig. 1 and Supplementary Fig. 1). ECL reaction takes place in the anodic cell compartment that, to this purpose, has a transparent window made of quartz suitable for collection of photons. Benzothiadiazole-triphenylamine (BTD-TPA) was chosen as the ECL chromophore for our investigations due to its high photoluminescence efficiency and stability in the film state (vide infra).

### Streaming potential measurements
Several parameters such as the pressure drop ($\Delta P$) between the electrodes, the zeta-potential of the channel walls and the solution conductivity are key factors that determine the buildup of streaming potential, according to the Smoluchowski's equation (Eq. (1))[21]:

$$E_{str} = \frac{\varepsilon_0 \varepsilon_r \zeta}{\eta K_L} \Delta P \tag{1}$$

where $\varepsilon_0$ is the electrical permittivity of vacuum, $\varepsilon_r$ is the relative permittivity of the solution, $\Delta P$ is the pressure drop (Pa), $\zeta$ is the zeta-potential of the channel walls (V), $\eta$ is the solution viscosity (Pa s) and $K_L$ is the solution conductivity ($\Omega^{-1}$ m$^{-1}$).

In our previous report, the cotton wool was tightly filled inside the PEEK tube to generate sufficient $\Delta P$ between the upstream and downstream chambers[22]. In this context, the system with lower $\Delta P$ and higher $E_{str}$ is ideal for practical application in a reaction device, and hence, we initially optimized the filler material to obtain the highest streaming potential value. We prepared two types of PEEK tubes (inner diameter: 1.0 mm) with different fillers (cotton wool or resin monolith). The phenolic resin monolith is known as a porous material with bicontinuous microchannels and a large specific interfacial area[32]. As shown in Fig. 2a, streaming potential measurements were carried out using the PEEK chambers connected through the PEEK tube containing one of the fillers. The two electrodes were connected with a voltmeter (V in Fig. 2a) to probe the solution potential difference generated inside the cell, meaning the streaming potential.

When a diluted electrolytic solution of acetonitrile (MeCN)/H$_2$O (3:1 v/v) containing 0.5 mM tetrabutylammonium hexafluorophosphate (Bu$_4$NPF$_6$) and 5 mM tri-$n$-propylamine (TPrA) was pumped at various flow rates, streaming potentials were generated and monitored

**(a) An ECL device powered by streaming potential**

**Bipolar Electrode (BPE)**

Inlet  PEEK tube with a filler  Outlet

Solution potential  $E_a$  $E_c$  $E_{str}$

Distance

**At the anodic side of the BPE**

BTD-TPA

BTD-TPA*

BTD-TPA$^{+\bullet}$  TPrA$^+$

TPrA  TPrA$^\bullet$

TPrA$^{+\bullet}$  H$^+$

BTD-TPA

**(b) Detection of amines based on the coreactant ECL system**

**Fig. 1 | Concept of this study. a** Graphical representation of the ECL microfluidic device powered by streaming potential-driven bipolar electrochemistry and **b** the corresponding strategy for detection of amines based on the coreactant ECL system. ECL: electrochemiluminescence. A: ammeter. PEEK: polyether ether ketone. $E_a$ and $E_c$: anodic and cathodic potential, respectively. $E_{str}$: streaming potential. BTD-TPA: benzothiadiazole-triphenylamine. TPrA: tri-$n$-propylamine.

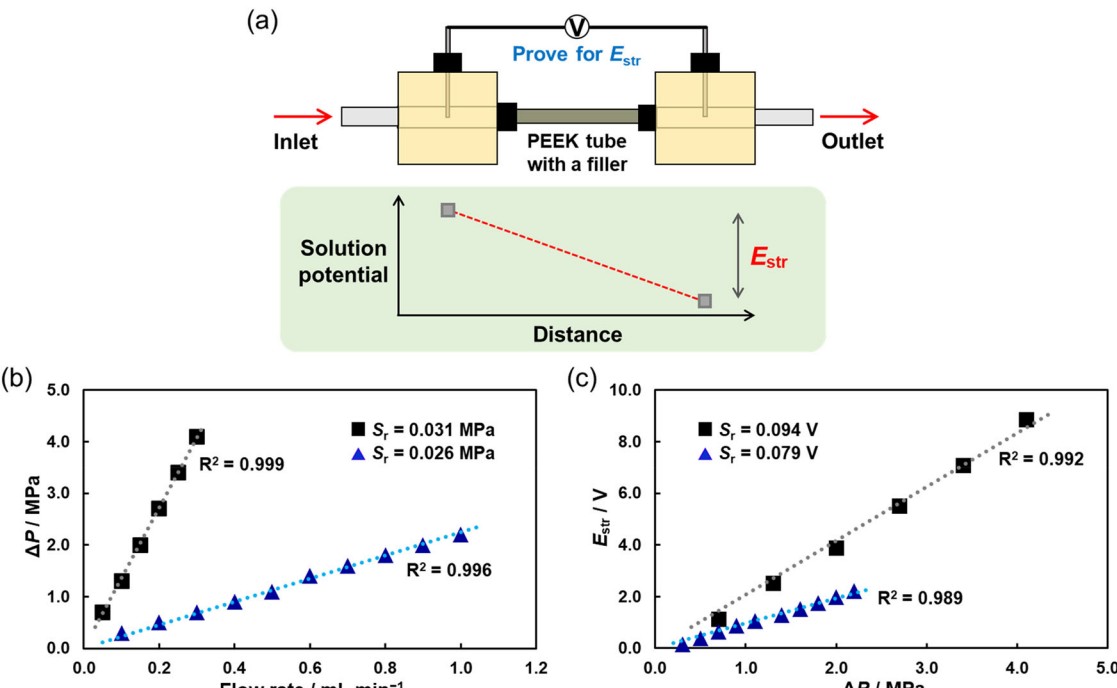

**Fig. 2 | Generation of streaming potential within the microfluidic device.**
**a** Illustration of the streaming potential measurement system using a voltmeter (V letter) between the two electrodes. Pressure drop ($\Delta P$) and streaming potential ($E_{str}$) were measured using the cotton wool filling or the phenolic resin monolith no. 1 filling during pumping of the MeCN/H$_2$O (3:1 v/v) solution containing 0.5 mM Bu$_4$NPF$_6$ and 5 mM TPrA. The relationship between **b** $\Delta P$ and flow rate and **c** $E_{str}$ and $\Delta P$. The black squares indicate the values measured with the phenolic resin monolith no. 1 as a filler, whereas the blue triangles represent the values measured with the cotton wool as a filler. Gray and light blue dotted lines represent the linear fitting. $S_r$ indicates the standard deviation of the regression. $R^2$ indicates the coefficient of determination.

with the corresponding discharge pressures, i.e., the pressure drop between the upstream and the downstream chamber ($\Delta P$ in Fig. 2b and c). For both filling materials, the $E_{str}$ value increases proportionally by increasing the flow rate and $\Delta P$ value. However, according to the results reported in Fig. 2c, the use of the phenolic resin monolith generated a steeper slope of $E_{str}/\Delta P$ than the use of the cotton wool, indicating that the phenolic resin monolith is a better filling material for the generation of larger $E_{str}$ with low $\Delta P$ in the solution system, presumably because of the large specific interfacial area generated from its bicontinuous channel structure. Moreover, since the phenolic OH group dissociates more easily compared to the cellulose OH group, the resin material probably generates a higher zeta potential to give higher $E_{str}$ in solution[33]. In this context, and also considering that electrochemical reactions are triggered under a flow condition, the device operation using low flow rates is ideal. According to these preliminary investigations, we concluded that the performance of the phenolic resin monolith is far superior to that of the cotton wool, and therefore, this material was adopted as the appropriate filler to be tightly inserted into the PEEK tube. For our convenience of use, we have employed two resin monoliths for subsequent investigations. Although these resin monoliths were prepared following the same reported procedure[32], they generated slightly different $E_{str}$ and $\Delta P$ values (Supplementary Fig. 2).

## Generation of ECL powered by streaming potential

We next explored the feasibility of the ECL reaction driven by the concept of streaming potential using the phenolic resin monolith as the optimal filler material. To our purpose, we opted for a solid-state ECL system, in which an ECL emitter is fixed onto the anode surface. This operation mode is suitable for our streaming potential equipment, because the electrolyte and the coreactant are solely pumped into the inlet channel to drive the ECL reaction at the anode. BTD-TPA

was chosen as the ECL emitter because of its remarkable ECL behavior in the solid-state, i.e., aggregation-induced ECL (AIECL) property, and for the relative stability of its radical cation[34]. Moreover, it also possesses a good ECL quantum efficiency and can generate films with a robust ECL behavior[35]. The BTD-TPA thin film deposited onto the Pt wire electrode was simply obtained by dipping the metallic wire into a chloroform solution of BTD-TPA once and allowing it to air-dry.

To detect the ECL signal of BTD-TPA at the upstream anode, the PEEK cell chamber was replaced with a transparent quartz cell chamber, where the light emission can be collected with a PMT placed closely and at a constant distance to the transparent cell chamber (Fig. 3a). This strategy is effective to our case, since the difference in the cell material did not affect the streaming potential behavior (Supplementary Table 1 and Supplementary Fig. 3). ECL experiments were carried out by pumping the same diluted electrolyte of the streaming potential measurements into the microfluidic system, i.e., the mixed solvent of MeCN and H$_2$O (3:1 v/v) containing Bu$_4$NPF$_6$ (0.5 mM) as a supporting electrolyte and TPrA (5 mM) as the coreactant. To drive the streaming potential ECL device, the two Pt wires were connected through a low-resistance ammeter, with the result that the pair of Pt wires behave as a split BPE (Fig. 3b). While the ECL reaction is expected to occur at the anodic side of the split BPE system, the reduction of water is presumably occurring at its cathodic side simultaneously, in order to maintain the electroneutrality of the split BPE system. Pumping of the fresh electrolytic solution started 60 seconds after the PMT was activated, in order to allow the stabilization of the background signal. From the homogeneous BTD-TPA coating at the Pt split BPE anode (Supplementary Figs. 4 and 5), the generation of a light emission was detected soon after feeding the device with the electrolytic solution, which immediately reached the maximum intensity, followed by a gradual decay (Fig. 3c, red line, Supplementary Table 2 and Supplementary Fig. 6). Indeed, when $E_{str}$

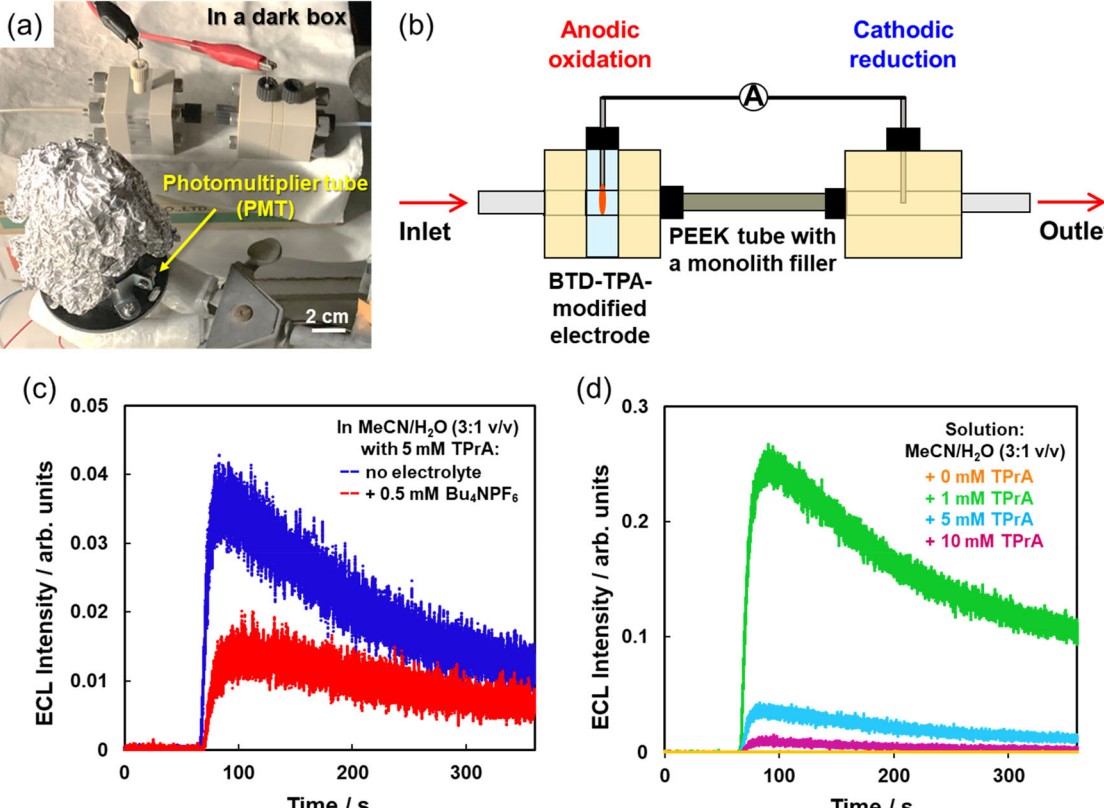

**Fig. 3 | ECL generation driven by streaming potential. a** Photograph of the streaming potential-driven ECL device with the quartz chamber and the PMT used as a light detector. The phenolic resin monolith was tightly filled inside the PEEK tube. **b** Schematic illustration of the device: the two electrodes were connected with an ammeter (A letter) to form a split BPE system to drive redox reactions. BTD-TPA was dip-coated onto the upstream electrode (anode). **c** A typical ECL emission profile generated during the flow of a MeCN/H$_2$O (3:1 v/v) solution containing 5 mM TPrA with 0.5 mM Bu$_4$NPF$_6$ (red line), or without supporting electrolyte (blue line), at the flow rate of 0.30 mL min$^{-1}$. Pumping of the solution started 60 s after the PMT was activated. **d** ECL emission profiles using an electrolyte-free MeCN/H$_2$O (3:1 v/v) solution containing different concentrations of TPrA. Phenolic resin monolith no. 1 was employed as the filling material in all these experiments.

reaches the suitable value to trigger the ECL reaction, both the chromophore and the coreactant, namely TPrA, are oxidized at the electrode surface. The oxidation ignites the cascade reactions that prompt the step increase in light intensity. In this first step, the TPrA located around the electrode is fully consumed by the oxidation process and, subsequently, fresh TPrA has to move toward the electrode to sustain the light generation. The mass transport process is governed by TPrA diffusion and forced convection induced by the flow of the electrolyte. Nevertheless, a gradual decay of the light emission is concomitantly observed, likely due to the physical deterioration of the BTD-TPA coating onto the electrode during the oxidation process. On the other hand, the continuous flow of fresh electrolyte can supply enough coreactant to sustain the light emission even for several minutes of device operation. In such a case, the process is only under convection control by the flowing electrolyte, while the diffusion layer thickness of TPrA is constant, giving the ECL emission a constant profile typical of a convection-controlled process[36].

As can be observed in Supplementary Table 2, the highest ECL intensity and the largest integrated ECL intensity were obtained by increasing the flow rate up to 0.30 mL min$^{-1}$. The $\Delta P$ had to be controlled within 5 MPa because larger pressures in the upstream chamber would cause the breakage of the quartz cell. Therefore, the maximum flow rate was limited to 0.30 mL min$^{-1}$. This condition is also appropriate for the analytical application of the device, because mass transport is controlled by a convection regime that provides a constant flow of fresh TPrA over the electrode surface, assuring ECL generation over time[36].

In these experimental conditions, the minimum $E_{str}$ required to trigger the ECL reaction was found to be 2.3 V (entry 2 in Supplementary Table 2), a value that is considerably larger to the threshold potential obtained by the voltage-sweep ECL measurements with the two-electrode system using the same electrolyte conditions (Supplementary Fig. 7).

As reported in the Supplementary Discussion, the relationship between the amount of BTD-TPA coated onto the electrode and the ECL intensity was investigated and it was found to be optimal using 20 mg mL$^{-1}$ of BTD-TPA solution (Supplementary Table 3 and Supplementary Fig. 8). In following experiments, the optimal composition of the mixed solvent was also investigated and it was determined that the ratio of MeCN and H$_2$O 3:1 v/v gave the highest ECL emission (Supplementary Table 4 and Supplementary Fig. 9). In our streaming potential device, the non-faradaic current (streaming current[37,38]) is counterbalanced by the ohmic current and the bipolar faradaic current[39], the latter derived from the redox processes developed at the split BPE system. In other words, the streaming current can carry part of all the current generated electrochemically. In addition, faradaic currents generated at electrodes embedded in fluidic channels can also generate electroosmotic flows in a manner that, coupled with convection, can generate effects complementary to streaming current and potential[37]. Hence, at this stage of our investigation, it is rather difficult to quantify the magnitude of the faradaic current flowing through the split BPE system.

In the previous report, it was found that streaming potential was generated even when flowing the pure MeCN solution through the

microchannel filled with cotton wool[22], because the very low solution conductivity contributes to the onset of streaming potential according to Smoluchowski's equation (Eq. (1))[37]. Therefore, we next investigated the streaming potential-ECL generation without any supporting electrolyte. Indeed, slightly higher $E_{str}$ values were observed in the electrolyte-free condition (Supplementary Table 5). Surprisingly, the flow of the electrolyte-free MeCN/H$_2$O (3:1 v/v) containing only TPrA (5 mM) showed significantly higher ECL intensity compared to the same solution containing 0.5 mM Bu$_4$NPF$_6$ electrolyte (Fig. 3c, blue line, and Supplementary Fig. 10). This effect may be ascribed to the generated cationic species of BTD-TPA and TPrA that, in solution without electrolyte, are naked, i.e., such radical cations are not stabilized by the anions derived from the supporting electrolyte and may be active for the ECL reaction. In particular, the low ionic strength of the electrolyte increases the ECL efficiency of TPrA, due to the increase of the deprotonation rate of the TPrA radical cation (TPrA$^{\bullet+}$) to generate the TPrA radical (TPrA$^{\bullet}$) species[40], which are both fundamental in the ECL generation process. Because the use of low concentrations of supporting electrolyte is also one of the main advantages of bipolar electrochemical systems[27], this result is remarkable, since the electrolyte-free solution is the optimal condition for our streaming potential-driven ECL device. Moreover, since the detection of the target species can be accomplished without modification of the sample's chemical environment and does not require any preconcentration or derivatization steps, the use of the device for sensing applications is also rather simple and practical.

Exploiting the electrolyte-free condition, we next examined the role of TPrA concentration in terms of device performance. As observed in Fig. 3d and Supplementary Fig. 11, 1 mM concentration exhibited the best ECL performances among the investigated concentrations of 0, 1, 5, and 10 mM. Considering the solid-state ECL mechanism of BTD-TPA[34], the concentration of the coreactant is a key factor, because the oxidation of both BTD-TPA and TPrA should occur at the anode with an appropriate balance. In fact, it is well known that aliphatic amines can quench the excited state of aromatic molecules[41], and, depending on coreactant concentration and applied potential, this may also occur in the frame of the ECL process[42]. Furthermore, we also investigated the effect of the pH, since it is also crucial in determining the ECL intensity in the coreactant ECL system of BTD-TPA. Our results showed that the maximum ECL intensity in the electrolyte-free solution was achieved at pH 10 (Supplementary Table 6), which is slightly higher than the value of 8 reported from other authors in buffered solution[34].

It is well known that the nature of the electrode, such as the material and its surface state, strongly influences the ECL emission process[43]. Generally, carbonaceous materials are good candidates for ECL applications because of their low cost, disposability and efficient TPrA oxidation[44,45]. However, metallic electrodes are usually preferred in commercial systems because of their stability and fast kinetics of electron transfer reactions[44,46]. Therefore, we next investigated the effect of the electrode material on the ECL performances of the device, including the generation of streaming potential. To this aim, we replaced the Pt anode of the split BPE system with different metals, such as gold (Au), silver (Ag), and copper (Cu). As shown in Supplementary Fig. 12, Pt showed the highest ECL intensity but with a pronounced decay, while Au showed a relatively stable, but lower ECL emission. Generally, the ECL performances of the Au material are superior to those of Pt, because passivation of the electrode surface, which is due to the generation of an oxide layer that negatively impacts TPrA oxidation, occurs at more positive potentials compared to Pt[43,47]. However, it is accepted that on the Pt electrode, ECL is mainly generated via the catalytic pathway[36,48], which does not involve direct TPrA oxidation but rather its homogeneous chemical oxidation by the oxidized chromophore, a fact that might explain the highest ECL intensity generated in the present case. ECL emission could not be detected using the less noble metals Ag and Cu, although the BTD-TPA films were firmly adhered to the electrodes

and no apparent metal dissolution was observed. Nevertheless, the use of Cu generated the highest current flowing through the split BPE system (Entry 4 in Supplementary Table 7), a proof that might be ascribed to copper oxidation that generally occurs at relatively mild potentials. Similarly, Ag oxidation, which also occurs at low anodic potentials, may be the predominant redox process at the split BPE anode, a fact that might prevent oxidation of the BTD-TPA chromophore, leading to a lack of light emission. However, the different nature of the electrodes did not significantly impact the magnitude of $E_{str}$ (Supplementary Table 7), since 0.2 V is the difference between the highest and the lowest $E_{str}$ values generated from the use of Cu and Pt, respectively.

Interestingly, we successfully recorded the ECL emission using a simple digital camera (ISO: 25600, f/2.8, exposure time: 30 s) during the flow of the electrolyte-free MeCN/H$_2$O (3:1 v/v) solution containing 1 mM TPrA (Fig. 4a, b, Supplementary Table 8 and Supplementary Fig. 13). The brightness of the recorded emission corresponded well to the ECL intensity profile obtained with the PMT in the same time range (Fig. 4c and Supplementary Fig. 13). When the solution was injected at higher flow rate (0.50 mL min$^{-1}$), stronger ECL intensity was observed and could be recorded by using the digital camera with a lower sensitivity (ISO: 12800, frame rate: 30 fps), even if the intensity of the light emission decreased to around the half in the first 100 s (Supplementary Fig. 13 and Supplementary Movie 1). ECL spectra were recorded with an optical fiber cable set close to the quartz cell and by flowing the solution at the rate of 0.50 mL min$^{-1}$. The obtained ECL spectrum showed a unimodal peak at $\lambda_{max}$ = 623 nm (Fig. 4d and Supplementary Fig. 14), which is consistent with the reported ECL and photoluminescence spectra of BTD-TPA in the solid state[35].

We additionally investigated the response and stability of the ECL signal respect to the on/off pumping of the solution into the streaming potential ECL device by simply injecting an electrolyte-free MeCN/H$_2$O (3:1 v/v) solution containing 1 mM TPrA with a regular interval flow of 10 s followed by a 10 s of pause, and by repeating this flow scheme 15 times. A periodic current pattern was observed, which corresponded to the pulsed feeding of the solution (Supplementary Table 9 and Supplementary Fig. 15). The overall profile of the ECL emission obtained during the on/off pumping of the solution was similar to the emission profile obtained during the continuous flow of the electrolyte, but with an accentuated and gradual decay. This effect might be ascribed to the physical deterioration of the BTD-TPA coating onto the electrode, as evidenced by the laser microscope images reported in Supplementary Fig. 16.

The results reported so far have been obtained by using a plunger pump to inject the solution into the streaming potential ECL device. However, the use of a syringe pump was also tested to feed the solution at a relatively high pressure. Using a smaller diameter syringe (2.5 mL), the flow of the solution at the rate of 0.10 mL min$^{-1}$ generated a similar streaming potential value compared to the case when a plunger pump was used. The volume of 2.5 mL was enough to monitor the streaming potential and to generate ECL, a feature that is ideal for small analytical devices since the detection of the target analyte must be accomplished in a low sample volume[19]. Furthermore, similar streaming potential values were generated even when the syringe was operated by hand, proving that the concept of a reliable electricity-less ECL system has been achieved.

## Detection of amines by such streaming potential ECL device

Based on all the above-reported results, we tested our device for the detection of amines in water-based solutions without the use of electricity. Our interest in amines was justified not only for their pivotal role as coreactants for ECL generation, but also because of their relevance as polluting agents for the environment and their dangerousness to human health. Indeed, both aliphatic and aromatic amines are widely used in many industries and, as a result of these applications, they are also dispersed in the atmosphere, water and soil, thereby

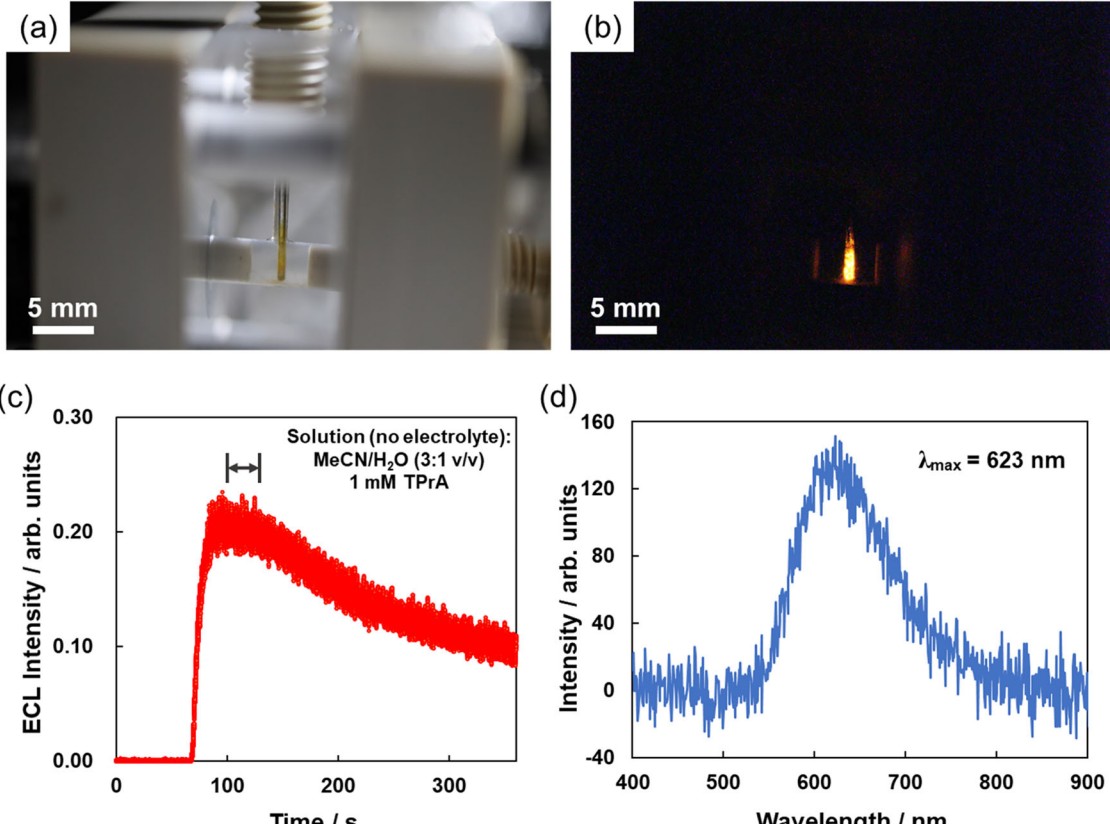

**Fig. 4 | ECL spectra and imaging of the BTD-TPA film coated onto the split BPE anode. a** Ambient light photograph of the BTD-TPA-coated electrode in the upstream chamber of the flow cell. **b** Correspondent ECL photograph taken with a digital camera using 30 s exposure (solution condition and time range are indicated in (**c**)). **c** ECL emission profile during the flow of an electrolyte-free MeCN/H$_2$O (3:1 v/v) solution in the presence of 1 mM TPrA at the flow rate of 0.30 mL min$^{-1}$ using the phenolic resin monolith no. 1 as the filling material. Pumping of the solution started 60 s after the PMT was activated. **d** ECL spectrum of a BTD-TPA film.

creating a prospect for human exposure[26,49]. They possess carcinogenic and mutagenic potential and can cause acute toxicity, including skin sensitization and eye irritation, in workers who are occupationally exposed[50]. If released in the aquatic environment, they can have toxic effects on aquatic ecosystems and can act as precursors for the formation of several potentially toxic species, including nitrosamines and nitramines[51]. Hence, their sensitive detection is a matter of primary importance.

As shown in Fig. 3d, ECL was not observed in the absence of TPrA, demonstrating its fundamental role as a coreactant for light generation. Hence, we evaluated the limit of detection (LOD) of our streaming ECL device for TPrA. ECL emission was monitored for solutions of MeCN/H$_2$O (3:1 v/v) containing different concentrations (1.0–0.01 mM) of TPrA at the flow rate of 0.30 mL min$^{-1}$ (Fig. 5a, Supplementary Table 10 and Supplementary Fig. 17). The maximum ECL intensity was observed for the solution containing 0.50 mM TPrA, while the emission decreased at lower concentrations. No light emission was detected for a solution containing 0.010 mM TPrA, concluding that the LOD of the device is 0.01 mM (1.0 × 10$^{-5}$ M). Other groups reported lower LODs for TPrA[52–54], suggesting that similar sensitivities have not yet been achieved in our system. Nevertheless, it is important to underline that the ECL detection in this device was performed without the use of supporting electrolytes or buffered solutions, differently from the other cases. This implies that the variation of the TPrA concentration in an unbuffered solution likely results in a deviation from the pH range suitable for the ECL reaction. A shift of the pH generally results in a variation of the acid–base equilibrium (p$K_a$ of TPrA = 10.4[55,56]), influencing the rate of the deprotonation step of

TPrA$^{\bullet+}$ that is the critical step in forming the strong reducing agent TPrA$^{\bullet}$[48,57]. However, while there still may be room for improvement in our device, the proposed technique offers a superior and more realistic approach for analysis, because the chemical modification of the sample environment is not required.

We additionally investigated the effect of the coreactant type on the ECL performances of our device. Because tertiary alkyl amines have been reported as the most efficient coreactants for ECL generation compared to secondary and primary amines[24,58], our choice focused on other tertiary amines with different substitution groups. Hence, 2-(dibutylamino)ethanol (DBAE) and triethanolamine (TEOA), having one and three hydroxyethyl groups, respectively, were selected, and their ECL behavior was tested against the TPrA standard (using 1 mM concentration for each amine). As shown in Fig. 5b, Supplementary Table 11 and Supplementary Fig. 18, DBAE showed a lower ECL intensity than TPrA at the flow rate of 0.30 mL min$^{-1}$, while ECL emission could not be detected when TEOA was used as coreactant. Since streaming potential and pressure drop values were similar among the three investigated amines, the different ECL behavior did not pertain to the different physical state of the device, but rather to the ECL mechanism. It has been reported that electron-withdrawing substituents on the amine close to the radical center of the molecule, such as the hydroxy group, tend to destabilize the radical intermediate, causing a decrease in the ECL activity[24,58–60]. Hence, this evidence might explain the progressive decrease of the ECL intensity by increasing the number of the electron-withdrawing groups on the amine, giving the order TPrA>DBAE > TEOA for our streaming potential device. Nevertheless, it is important to underline that in our system, the substitution effect may

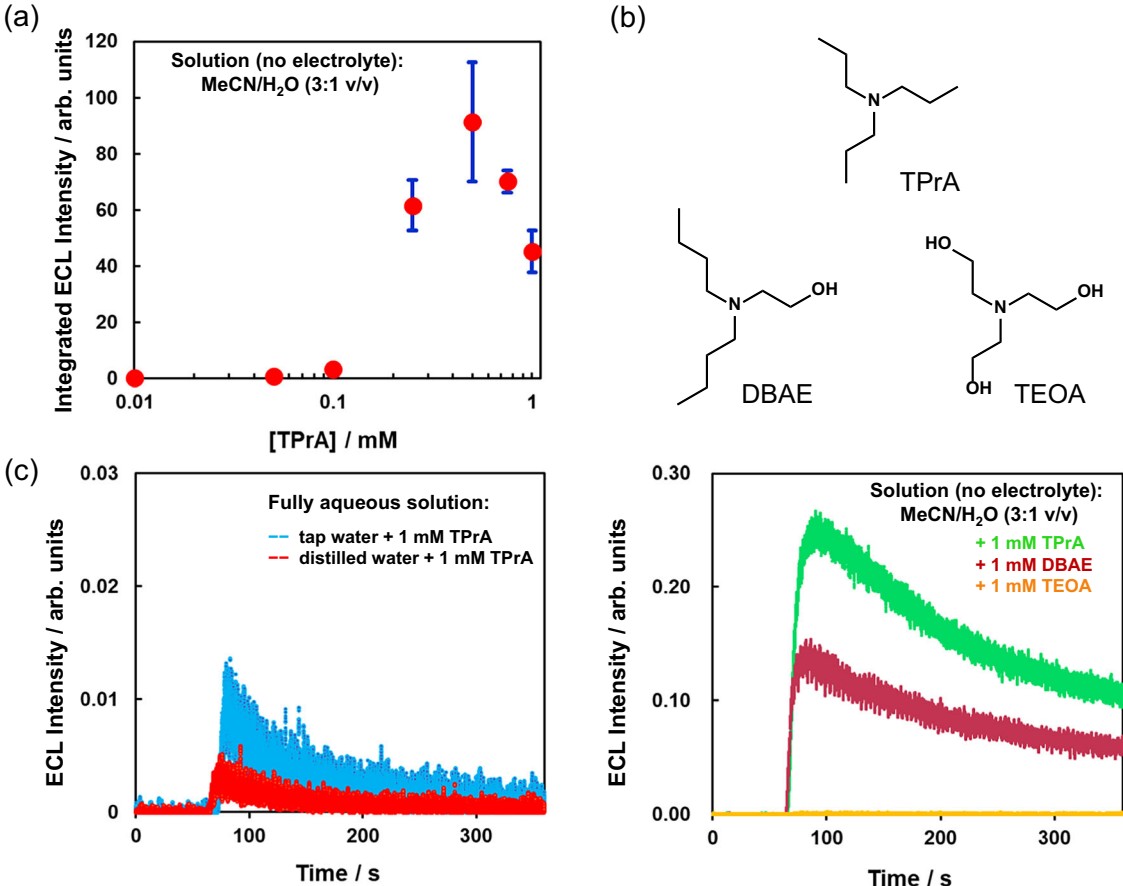

**Fig. 5 | ECL detection of amines in aqueous solutions using the streaming potential device. a** Dependence of the ECL intensity on the concentration of TPrA using an electrolyte-free MeCN/H$_2$O (3:1 v/v) solution at the flow rate of 0.30 mL min$^{-1}$. **b** ECL emission profiles obtained during the flow of a MeCN/H$_2$O (3:1 v/v) solution containing different amines (1 mM each) at the flow rate of 0.30 mL min$^{-1}$. **c** ECL emission profiles obtained using a fully aqueous system as the electrolyte: distilled water or tap water solutions containing 1 mM TPrA were injected into the flow cell at the flow rate of 0.30 mL min$^{-1}$. All measurements were performed using the resin monolith no. 1 as the filling material. Error bar shows the standard deviation ($n = 3$).

not be the determining factor for explaining the observed trend, since the experimental conditions used to operate the device are far from the common ones employed in ECL analysis. Even though such conditions are ideal for the buildup of $E_{str}$ values sufficiently large to drive redox processes on the Pt electrodes, they also undoubtedly affect the efficiency of the coreactant. For instance, it has been reported that low concentrations of DBAE produce higher ECL intensities compared to TPrA in the same concentration range[61]. Indeed, when the streaming potential device was operated at low flow rates up to 0.20 mL min$^{-1}$, 1 mM DBAE generated higher ECL intensity compared to 1 mM TPrA (Supplementary Fig. 18a–d and integrated ECL intensity values in Supplementary Table 11). However, at high flow rates, the trend reversed, and TPrA generated the highest ECL emission. Since the radical intermediate of DBAE is much more unstable than that of TPrA[62], the high flow rate presumably decreases the efficiency of this coreactant in our system.

We also explored the use of our device for the ECL detection of several aromatic amines, since they are chemicals of concern for the environment, and can cause acute toxicity and bioaccumulation in humans and ecosystems[49–51]. Aniline, *N*-methylaniline and *N,N*-dimethylaniline were investigated as coreactants for ECL generation in the streaming potential device by flowing an electrolyte-free MeCN/H$_2$O (3:1 v/v) solution containing 1 mM of each target aromatic amine at the flow rate of 0.30 mL min$^{-1}$. As summarized in Supplementary Table 12, all aromatic amines generated similar $\Delta P$, $E_{str}$, and $i$ values, indicating that the different aromatic structures did not influence the physical

state and operation of the device. Interestingly, the conductivity values of these solutions were considerably lower compared to the values obtained using aliphatic amines as coreactants (comparison of conductivity data between Supplementary Tables 11 and 12), a fact that is, however, beneficial for the buildup of streaming potential in our system. Despite these positive findings, no ECL emission was detected for all three investigated aromatic amines. This result is not surprising, since it is well known that aromatic amines do not show any ECL activity. Because the aromatic structure of the amine can delocalize the positive charge formed after electro-oxidation through resonance, the intermediate radical species tend to be excessively stabilized and, hence, less reactive towards the oxidized chromophore, hindering the ECL reaction[60,63]. It should also be noted that aromatic amines, such as anilines, have been reported to behave as quenchers of the ECL emission[64], likely due to an energy transfer mechanism from the emitting excited state to electro-oxidation products of the amine[65].

**Streaming potential ECL generation in fully aqueous systems**

Considering a more practical and realistic application of our ECL streaming device for analytical purposes, we investigated a fully aqueous system containing a low concentrated coreactant as the electrolyte. In particular, distilled water containing 1 mM of each coreactant was used for the experiment. When the solutions were pumped into the cell at the flow rate of 0.30 mL min$^{-1}$, sufficient $E_{str}$ values were observed but the ECL intensities were rather low in every case compared to the results obtained using the mixed solvent of

MeCN/H$_2$O (3:1 v/v) system (Fig. 5c, Supplementary Table 13 and Supplementary Fig. 19). In particular, ECL emission could be detected using the TEOA-containing solution, differently from the previously observed results obtained with the mixed solvent, even if its ECL emission profile exhibited the fastest decay and thus the smallest integrated ECL intensity among the three investigated amines. This result may be ascribed to the lower solubility of TPrA and DBAE in water compared to TEOA, as well as to the hydrophobic character of the BTD-TPA film, which is probably not suitable for the coreactant ECL reaction in the same solvent.

We also tested the performance of the ECL streaming device by using tap water (Yokohama city Kawai purification plant) as the solvent to solubilize the coreactant. The $E_{str}$ value observed using tap water was considerably lower compared to the use of distilled water, presumably because tap water includes minerals and presents higher ionic conductivity compared to distilled water (comparison between Supplementary Tables 13 and 14), which negatively affects the generation of streaming potential. Nevertheless, higher ECL intensity was detected using tap water, likely because some dissolved minerals, such as carbonate[66,67], may buffer the protons in proximity of the electrode surface, leading to a faster TPrA deprotonation (Fig. 5c and Supplementary Fig. 20). In addition, considering the relatively higher $E_{str}$ and current values generated using distilled water compared to tap water (comparison between $E_{str}$ and $i$ data in Supplementary Tables 13 and 14), it is not possible to exclude that the water oxidation process may occur at the anode in the former case, and, hence, may compete with the ECL process causing a decrease of the light intensity.

We lastly evaluated the anti-interference capabilities of the streaming potential device for sensing amines. We initially performed this study by flowing in the device the standard electrolyte, namely the electrolyte-free MeCN/H$_2$O (3:1 v/v) solution, containing a mixture of multiple amines, generally TPrA and another aliphatic or aromatic amine, at the flow rate of 0.30 mL min$^{-1}$. The ECL detection of such multiple coreactant systems was then compared with the ECL signal of the standard solution containing only 1 mM TPrA. When the flowing electrolyte contained in the same concentration TPrA and another aliphatic amine, such as DBAE or TEOA, the resulting ECL signals were considerably lower compared to the solution containing only TPrA, even though $E_{str}$, currents and conductivity values were similar among the cases (Entries 1–3 in Supplementary Table 15 and Supplementary Fig. 21a). If an aromatic amine, like aniline, coexisted in the solution with TPrA, no ECL emission was observed, owing to the quenching effect of the aromatic species towards the emitting excited state[65] (Entry 4 in Supplementary Table 15 and yellow line in Supplementary Fig. 21a). However, an enhancement of the ECL emission was observed by decreasing the concentration of both TPrA and DBAE in the flowing solution (comparison between Entries 2, 5 and 6 in Supplementary Table 15 and ECL curves in Supplementary Fig. 21b). Because a decrease of the amine concentrations gives rise to a decrease of the solution conductivities and generates larger pressure drops, this is likely advantageous for the retention of streaming potential in the device. However, even if low solution conductivity was also observed by increasing the concentration of TPrA from 1 to 2 mM, the drastic decrease of the ECL intensity (Entry 7 in Supplementary Table 15 and blue line in Supplementary Fig. 21a) highlights the complex balance of conditions suitable for the generation of both streaming potential and ECL.

We finally evaluated the anti-interference performances of the device using tap water with the addition of 1 mM TPrA as the flowing solution. We also studied the effect of other analytes in the solution, such as NaCl, that in 0.5 M concentration simulates the high salinity typical of sea water. Because of the relatively high ionic conductivity of these solutions, the device was operated at the maximum flow rate of 1.0 mL min$^{-1}$ in order to achieve satisfactory $E_{str}$ values. As summarized in Supplementary Table 16 and Supplementary Fig. 22, ECL emission could be detected in tap water containing only TPrA, but the flow of this solution generated a higher pressure drop compared to the electrolyte-free MeCN/H$_2$O (3:1 v/v) solution containing the same TPrA concentration (comparison of $\Delta P$ values in Entries 1 in Supplementary Tables 15 and 16). The addition of NaCl completely suppressed the generation of streaming potential, owing to the extremely high ionic conductivity of the solution according to Smoluchowski's equation (Eq. (1)), and ECL reaction could not be initiated in this case.

## Discussion

We have reported an original type of microfluidic device with integrated ECL detection based on the concept of streaming potential-driven bipolar electrochemistry and applied it for the detection of amines in water by exploiting their role as coreactants for ECL generation. The device can be operated without the use of an external electrical power source, because a pressure drop established through a microchannel with a filling material can generate a sufficient potential difference, namely streaming potential, to drive faradaic reactions at the surface of a split BPE. Indeed, a simple syringe operated manually is sufficient to establish enough pressure drop suitable for redox reactions. Notably, the ECL reaction of an organic chromophore coated onto the electrode was successfully demonstrated in such kind of device by using either electrolyte-free and fully aqueous solutions containing only a low-concentrated coreactant. These results are significant because they demonstrate the possibility to drive redox reactions in the absence of electrolytes or without the use of buffered solutions, because the low ionic conductivity of these solutions promotes the buildup of streaming potential in the device. The application of this microfluidic system for sensing amines using such conditions showed high reproducibility and satisfactory limits of detection for TPrA. Remarkably, the operation of the device in a fully aqueous solution is also rather simple and practical, since the streaming potential-driven ECL system can operate reliably without any sample pretreatment step. We believe that our prototype may represent an innovative class of low-cost and portable analytical electrochemical devices that can be employed by using the electrical power of nature. Indeed, our future envision is that, once this technology has advanced and become more solid, a continuous natural water flow, e.g., a river flow, can be exploited to produce the necessary electrical energy to run the device. This would realize an electroanalytical system with integrated ECL detection that could function without anthropic-derived electricity, with potential utility in emergency situations or in areas with limited access to electricity.

## Methods

### General considerations

All reagents and solvents were obtained from commercial sources and used without further purification. Platinum (Pt) wires were purchased from Nilaco Corporation. Streaming potentials between the two cell chambers and the currents flowing through a split bipolar electrode (split BPE) were measured with a Sanwa PC773 multimeter connected to a PC Link 7 software. Flow cells were custom-made with polyether ether ketone (PEEK). Electrolytes were pumped into the cells with a Shimadzu LC 20 AD plunger pump and by monitoring its discharge pressure. Voltage–current curve measurements were conducted with a two-electrode system equipped with Pt electrodes and recorded with a Metrohm PGSTAT101 potentiostat. Electrochemiluminescence (ECL) was detected with a photomultiplier tube (PMT, Hamamatsu R928HA) controlled by the use of a Metrohm PGSTAT101 potentiostat, or with a Canon EOS Kiss X10 digital camera. ECL spectra were obtained with an Ocean Insight QEPro high-performance spectrometer via an optical fiber cable positioned in front of the transparent cell chamber. The pH of the electrolytic solutions was measured with a Horiba LAQUA pH meter.

## Microfluidic electrochemical cell

A custom-made PEEK cell, composed of two chambers, each one accommodating a platinum (Pt) wire (Ø = 0.6 mm) and a PEEK microtube (outer diameter: 1/16 inch) connecting both chambers, was used as a flow reactor to drive electrochemical reactions by exploiting the concept of streaming potential-driven bipolar electrochemistry. The Pt wires were inserted into the chambers by using polytetrafluoroethylene (PTFE) tubes and were connected with a multimeter (voltmeter or ammeter) (Supplementary Fig. 1). The electrolytic solution was fed by a plunger pump with a constant feed rate.

## Streaming potential measurements

The value of the pressure displayed by the pump reflected the discharge pressure for the fluid. In our device, this pressure was gradually reduced across the microchannel to reach the atmospheric pressure in the downstream chamber, implying that $\Delta P$ could be monitored and tuned with the feeding pump by changing the flow rate. The two Pt wires inserted into the chambers were connected with a voltmeter to detect the streaming potential ($E_{str}$) generated between the chambers by a flow of the electrolyte. To generate such a pressure drop between the upstream and downstream chambers, cotton wool (20 mg) or phenolic resin monolith[32] was tightly filled into the PEEK tube (inner diameter: 1.00 mm). When the electrolytic solution of acetonitrile (MeCN)/water (3:1 v/v) containing 0.5 mM tetrabutylammonium hexafluorophosphate ($Bu_4NPF_6$) and 5 mM tri-*n*-propylamine (TPrA) was pumped at various flow rates, $E_{str}$ values were generated and monitored with the corresponding pressure drop ($\Delta P$) between the upstream and downstream chambers (Fig. 2a). For the ECL experiments, the use of a transparent chamber was required to detect the photons generated from the emitter during the ECL reaction at the upstream anode; hence, the PEEK cell chamber was replaced with a transparent quartz cell chamber. The difference in the cell material did not affect the generation of streaming potential (Supplementary Table 1 and Supplementary Fig. 3).

## BTD-TPA coating onto the electrode

Benzothiadiazole-triphenylamine (BTD-TPA) was prepared according to the literature[35] and used as an ECL chromophore. A Pt wire (Ø = 0.6 mm) was coated with BTD-TPA by dipping the metallic wire into a chloroform solution of BTD-TPA once, and then dried in the air. A freshly prepared BTD-TPA-modified Pt wire was used for each ECL experiment. An Olympus OLS4100 3D laser microscope was employed for the observation and evaluation of the BTD-TPA coating on the Pt wire[68].

## Generation of ECL through streaming potential in the microfluidic cell

To drive the streaming potential ECL device, the two Pt wires inserted into the chambers were connected with a low-resistance ammeter, with the result that the pair of Pt wires behaves as a split BPE[29–31]. Using this configuration, the upstream electrode was dip-coated with BTD-TPA as the ECL emitter and positioned in the transparent quartz cell chamber. An acetonitrile (MeCN)/$H_2O$ (3:1 v/v) solution containing low concentrated TPrA and supporting electrolyte ($Bu_4NPF_6$) was used as the electrolyte and pumped into the cell using a plunger pump. A PMT, positioned close and at a constant distance from the transparent quartz cell chamber, was used to record the ECL emission generated from the BTD-TPA coating on the anodic side of the split BPE.

## Data smoothing for ECL intensity detected by PMT

Data smoothing was carried out with the moving average (MA) method. MA is the mean over the last $i$ entries of a data-set containing $n$ entries, and is denoted as Eq. (2):

$$MA_i = \frac{x_n + x_{n-1} + x_{n-2} + \cdots + x_{n-(i+1)}}{i} \tag{2}$$

where $x_1, x_2, \ldots, x_n$ are the data-points. In this study, the scan speed was $0.005\,s^{-1}$ and the smoothing process was performed using MA ($i = 20$) every 0.1 s. After smoothing, the maximum ECL intensity was determined. In addition, integrated ECL intensity was estimated using sums of areas of rectangles to approximate the area under a curve (Riemann sum).

## Reporting summary

Further information on research design is available in the Nature Portfolio Reporting Summary linked to this article.

## Data availability

The data that support the findings of this study are available from the corresponding authors upon request. Source data are provided with this paper.

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

## Acknowledgements

This work was supported by the FOREST Program (JPMJFR211G) of the JST, a Kakenhi Grant-in-Aid (JP22K14708) from JSPS, and a Grant-in-Aid for Transformative Research Areas (A) Green Catalysis Science for Renovating Transformation of Carbon-Based Resources (Green Catalysis Science) (MEXT KAKENHI Grants JP23H04914) from MEXT. We also thank a Support for Tokyo Tech Advanced Researchers (STAR) grant funded by the Tokyo Institute of Technology Fund (Tokyo Tech Fund).

## Author contributions

These authors contributed equally: Rintaro Suzuki, Suguru Iwai and Ryota Kirino. R.S. and S. Iwai performed experiments on ECL generation and optimization of the device, performed detection of aliphatic amines and carried out data elaboration. R.K. performed experiments on anti-interference performances of the device and ECL detection of aromatic amines and carried out data elaboration. K.S. supervised the project and discussed the results. M.K. performed preliminary experiments on ECL generation with the device and offered technical support throughout the study. G.H., N.I., and K.M. prepared the resin monolith fillers for optimization of streaming potential. K.T. conceptualized the study and discussed the results. E.V. supervised the project, conceived the experiments and discussed the results, managed funding acquisition and wrote the original draft of the manuscript. S. Inagi conceptualized the study, conceived the experiments and discussed the results, man-aged funding acquisition and performed review and editing of the manuscript.

## Competing interests

The authors declare no competing interests.
