## [Transparent Peer Review file · Nature Communications]

An electrochemiluminescence device powered by streaming potential for the detection of amines in flowing solution

Corresponding Author: Professor Shinsuke Inagi

Version 0:

Reviewer comments:

Reviewer #1

(Remarks to the Author)

See the file attached.

Reviewer #2

(Remarks to the Author)

The manuscript presents ECL measurements using a (split) bipolar electrode. The experiments are described in detail, including the rationalization of several design choices. I however believe that the motivations behind this work are overstated, in particular:

1. The authors emphasize that a major benefit of the device is that it can be operated without external electrical power source, for example for use “in regions that do not have ready access to electricity”. I believe that this aspect is severely overstated since a simple battery would be sufficient to drive the ECL experiment. This is much simpler and practical than employing an external pump to induce a streaming potential through a microfluidic cell. I understand that the basic phenomenon can be produced by hand with a syringe, but presumably the flow rate will not be accurate enough for analytical applications in that case.
2. Furthermore, direct electrical actuation would be far more energy efficient (another purported advantage of the approach) as it would eliminate the very significant losses from viscous drag in the fluid. Based on the numbers in the manuscript, the mechanical power needed to drive the device is ~20 mW, about 4 orders of magnitude higher than the power consumed by the faradaic process (order 1 μ W). This is highly inefficient.
3. Wider applications are mentioned, for example in the realm of diagnostics. However, the experiments presented here are performed at very low electrolyte concentrations, ~100x smaller than physiological salt concentrations. Since the streaming potential is suppressed at high ionic strength, it is far from clear that the approach could be deployed to this broader class of systems. Neutral pH could also lead to a decrease of the zeta potential of the channel medium, further decreasing the streaming potential.
4. Although the authors claim to have performed “an extensive investigation of all the experimental parameters in play”, there is an important omission. During operation, the two electrodes are short-circuited via an ammeter. Were the electrodes close to non-polarizable, no significant streaming potential could develop. Instead, a streaming current would exist that would loop back through the ammeter. The fact that a streaming potential develops thus depends on the size and nature of the electrodes, such that some degree of polarization is possible. This is however not addressed in the manuscript, making it practically impossible to generalize the experimental results shown here to a more general situation.

Some additional points to improve the manuscript include:

5. The statement that the device “can be operated without the need of any type of energy” (abstract, page 2) is clearly incorrect as mechanical energy powers the experiment. Statements elsewhere in the manuscript are more carefully worded, and it would be preferable to be more precise in the abstract as well.
6. It is unclear what the pressure drop is in the inlet and outlet tubing, and hence the statement “the discharge pressure displayed by the pump is the internal pressure of the upstream chamber” could be further justified.
7. The term “ECL integral area” is not very intuitive, this terminology could be adjusted for clarity.
8. The text is lucidly written and easy to follow. There are however a few language mistakes, for example “an extensive investigations” (page 2), “whit” (page 7), and “did not significantly changed” (page 12).

Reviewer #3

(Remarks to the Author)

In this article, the authors report a fluidic device that accomplishes electrochemiluminescence (ECL) detection for amines in a flowing solution, harnessing the concept of streaming potential-driven bipolar electrochemistry. Pressure driven flow through a microchannel filled with a porous material generates a streaming potential, which in turn, drives ECL reactions at the anode of a bipolar electrode (BPE) that spans the channel. Experimental parameters including properties of the filler material, flow rate, electrolyte composition, chromophore's loading onto the electrode, and the type of co-reactant are investigated. The device achieves a limit of detection (LOD) of 0.025 mM for tripropylamine (TPrA) in MeCN/H₂O (3:1 v/v) solution without requiring supporting electrolytes or buffered solutions. The ECL intensities in fully aqueous systems are low compared to the results obtained using the mixed solvent system.

This work is important because it addresses challenges to the practical application of pressure-driven bipolar electrochemistry including 1) incorporation of ECL reagents (luminophore) into a solid phase on the BPE anode, 2) a need for larger streaming potentials, and 3) removal of the need for electrolytes or buffered solutions. The device developed and characterized in this study operates without external power, is low-cost, and is portable — ideal for sensing applications in emergencies or regions with limited access to electricity.

This manuscript is well-structured, and the figure captions and legends are very clear. However, certain sections of the article would benefit from additional discussion to better explain the experimental results and fundamental mechanisms. Further, the potential for practical application is not argued strongly enough.

Major Comments:

1. The data presented here is primarily collected in 3:1 MeCN:H₂O as the solvent to enhance the ECL signal. While signal was obtained in distilled water and tap water, even 1 mM TPrA (model analyte) produced a weak signal. Under what conditions, theoretically, could the reported system be useful for sensing amines at environmentally or clinically relevant concentrations?
2. Page 8, line 121. The authors indicate that the current flowing through the BPE is the sum of the faradaic current and the streaming current. However, in this system the streaming current and the faradaic current flow in series. The streaming current is the sum of the faradaic current and the charging current (non-faradaic). Further, on page 8, line 210, the authors indicate that it is difficult to estimate the current efficiency (percent of current being used to drive ECL) because of the mixed contribution of faradaic and streaming current in this system. The reviewer's understanding is that it is faradaic and charging current that are additive (not streaming current), and therefore, the current efficiency can be obtained following the decay of charging current. Further, if needed, the authors can operate this same device in an electrolytic mode (e.g. amperometric or voltammetric) to evaluate efficiency.
3. Page 13, line 321. The authors indicate that the low sensitivity of this system may be caused by lack of a buffer (to control deprotonation of the co-reactant). The authors may consider other mechanisms for the low sensitivity. It is probable that the TPrA that is oxidized at the BPE anode is being swept away by the flowing solution prior to subsequent reaction with the luminophore. Potential evidence of this effect would be a decay in the ECL intensity at very high flow rates. If this is the case, then the signal would be enhanced by positioning the electrodes in stagnant fluid outside of the direct line of fluid flow. Along the same line of thought, differences in ECL intensity among the three co-reactants tested may have an underlying kinetic mechanism. Could the authors comment on the potential for TPrA radical to be swept away prior to reacting with the luminophore/coating?
4. Page 14, line 358. Since there are low or no supporting electrolytes in the fully aqueous systems, the solution conductivity is very low. Is there a possibility that the low ECL signal is limited by not having enough charge carriers in the solution? Under certain conditions, is it possible that the streaming current limits the current through the entire circuit (including BPE current)?

Comments:

1. Page 2, line 40. The statement saying that the device can be operated without the need of any type of energy is imprecise.
2. Page 2. The sensitivity of the device and LODs of the amines are not mentioned in the abstract.
3. Missing details in both the article and the SI on how the area and thickness of the BTD-TPA film deposited on the Pt wire electrode are controlled.
4. S10. What is the difference between the 6 runs shown in Figure S4 (b) and (c)?

5. Page 7, line 192. Could the authors provide an explanation as to why the ECL signal reaches the maximum intensity immediately and then gradually decays?
6. Page 9, Figure 3 (c) and (e). These two figures appear to be slightly repetitive; the author can consider deleting Figure 3 (c) and combining the captions for the two figures.
7. Page 9, Figure 3 (e) shows that the ECL signal of the electrolyte-free solution is higher than that of the same solution with added electrolyte. How do the corresponding current signals compare?
8. Page 10, line 228. This statement needs to be clearer. The low solution conductivity doesn't contribute to onset, but to magnitude.
9. Page 10, line 249 and line 254. What's the authors' explanation for why a concentration of 1.0 mM TPrA and pH 10 exhibit the best ECL performance?
10. S19, Table S7. The solution conditions are the same, why is there a significant difference between the ΔP and Estr of entry 1 and entry 2? Both ISO and exposure time are set differently for the still imaging and video, the results should be incomparable.
11. S20, Figure S9 (i). How is this plot generated? Given that the data from the digital camera's still imaging should yield one ECL signal every 30 seconds, the methodology for producing this plot requires clarification.
12. Page 12, line 287-289. How stable is the BTD-TPA coating? How long can it last and is it reusable?
13. S23, Figure S11 (g) and (h). SEM images can be more convincing here.
14. Page 15, line 398. Instead of "self-powered", a better way to describe this system may be no electrical power or no external power supply, or hydro-powered or pressure driven.
15. This study only investigates TPrA, DBAE, and TEOA as co-reactants, what about other amines with clinical relevance, such as aromatic amines or trimethylamine? How much lower is the signal expected to be when these classes of amine are employed?

Reviewer #4

(Remarks to the Author)

Reviewer #5

(Remarks to the Author)

This study presents a novel micro-fluidic device utilizing bipolar ECL detection driven by streaming potential. The device is a simple and low-cost analytical tool that operates without external energy sources. While this research is innovative and appealing, the evaluation of the sensing system's anti-interference capabilities is limited. I recommend acceptance of the article following more comprehensive investigations into anti-interference performance using actual samples.

Version 1:

Reviewer comments:

Reviewer #1

(Remarks to the Author)

The authors have addressed my comments adequately. All the very best to the authors!

Reviewer #5

(Remarks to the Author)

In the revised manuscript, the authors conducted additional research on the anti-interface performance of the device. Experimental results indicate that the electrolyte significantly affects detection performance, and the fabricated device is only suitable for use in low salt concentration systems. I recommend that the manuscript be accepted in its current revision.

Reviewer #6

(Remarks to the Author)

The manuscript describes a microfluidic electrochemiluminescence (ECL) platform utilizing streaming potential for amine detection, with an emphasis on energy-efficient operation and portability. While the concept of eliminating external electricity through streaming potential-driven bipolar electrochemistry is interesting, the level of insight and innovation is somewhat lacking, particularly in comparison to the standards expected for publication in Nature Communications. I do not think that introducing a microfluidic system, consisting of a microchannel and pump, to replace the electrochemical workstation, is necessary. The workstation in fact can be even discarded by using an external voltage such as a battery. Portable ECL devices powered by batteries are already a mature and well-established technology. In fact, replacing a battery with a pump introduces unnecessary complexity, potentially complicating a relatively straightforward issue. Additionally, the authors repeatedly emphasize that the device can operate without electricity. I wonder whether the pump used for driving the fluid requires electricity. If the pump consumes electricity, it is not entirely accurate to claim that this system is truly "electricity-free". If it can operate manually, how can the reproducibility and accuracy of the measurements be ensured. Furthermore, as

depicted in the manuscript, the streaming potential is affected by multiple factors, including flow rate, conductivity and pH of liquid. When applied to real-world samples, these variables could significantly impact the analytical performance of the device, potentially leading to inconsistent or unreliable results. On the other hand, the detection of coreactant-type analytes via ECL has been extensively reported in prior work (e.g., *Analyst*, 2021, 146, 5198; *ChemElectroChem* 2020, 7, 1207; *Chem. Commun.*, 2016, 52, 12845). Furthermore, the inherent selectivity limitations of ECL for amine detection (a well-recognized challenge in the field) are neither partially addressed nor mitigated in this work. Given these concerns, I do not recommend this manuscript for publication in *Nature Communications*.

General comments:

1. I disagree with the sentence "It should also be important to underline that part of the great success of ECL is also due to the simple and relatively inexpensive equipment necessary to perform an ECL assay". Compared with photoluminescence and electrochemical techniques, ECL instruments are generally more complex. And the most successful applications of ECL, namely the automated ECL immunoassays, rely on sophisticated and advanced equipment rather than simple or inexpensive setups.
2. As author mentioned in Introduction: "the precise spatial and temporal control over the light emission reaction guarantees high selectivity, meaning that the detection of the target analyte can even be achieved in complex matrix such as urine or blood". However, the good spatiotemporal control of ECL is not directly related to high detection selectivity. While it aids in enhancing detection sensitivity and enables remote ECL imaging of large objects, such as single cells, detection selectivity arises from specific recognition events—such as the binding between antigen and antibody, the selective catalytic reaction between enzyme and substrate, or complementary base pairing between nucleic acid strands.
3. I also disagree with the sentence, "However, metallic electrodes are usually preferred because of their stability and fast kinetics of electron transfer reactions". The metallic electrodes are more susceptible to electrochemical oxidation compared with carbon-based or metal oxide electrodes, which often exhibit superior stability under oxidative conditions. That is why in commercial immunoanalyzer the platinum working electrode must be repeatedly renewed prior to ECL detection.
4. For ECL detection of analytes, the requirements of electrolytes/buffer solution, sample pretreatments, and chemical modification of the analyte environment are not necessarily drawbacks, as they do not compromise the quantitative accuracy of the results. In the case of detecting amines in pure samples, such as tap water, sample pretreatment is also not required in conventional portable ECL devices.

Specific comments:

1. How do the authors keep the batch-to-batch reproducibility of luminophore coating on the electrode? Please include characterization data such as electron microscopy (e.g., SEM or TEM) to support the consistency and uniformity of the coating.
2. In the manuscript, it is mentioned that the weakening of light intensity is due to the physical or chemical deterioration of the BTD-TPA. While images of physical detachment have been provided later in the text, further evidence is needed to substantiate the chemical factors contributing to this deterioration. Please include additional experimental data or analysis to support the chemical degradation hypothesis.
3. In Supplementary Table 2, the authors described that the minimum voltage required to trigger ECL is 0.23 V. However, this may not be accurate, as the triggering voltage is likely in between 1.0 V and 2.3 V. Please revise the statement or provide additional data to validate this claim.
4. In Supplementary Table 2, the E_{str} value for entry 4 is significantly higher than that for entry 3, yet there is no noticeable difference in their light intensities. Please provide further explanation or analysis to clarify this discrepancy.
5. The reaction mechanism between BTD-TPA and TPrA should be discussed in more detail. As shown in Figures 3e and 5a, why does the ECL intensity initially increase and then decrease with the increase of TPrA concentration? For instance, in Table 6, at a flow rate of 0.3 mL/min, the E_{str} and i values for 5 mM TPrA are higher than those for 1 mM TPrA, yet the light intensity decreases to 1/6.
6. As shown in Figures 5a, when the concentration of TPrA exceeds 0.5 mM, the ECL intensity decreases, suggesting that both low and high concentrations of TPrA may produce similar ECL signals. In this case, how do authors propose achieving accurate quantification of TPrA in the sample solution?
7. In Supplementary Figure 5, panel (c) exceeds the maximum range. Please redraw the figure to ensure that the data falls within the appropriate range.
8. For Ag and Cu, the E_{str} and i values are generally consistent with those of Pt and Au; however, no detectable light intensity is observed. I wonder whether the luminophore does not firmly adhere to the electrode. Or other reasons?
9. What is the concentration range of TPrA in aquatic ecosystems?
10. How did the authors ensure the selectivity of the device for TPrA detection?
11. Error bar is missing in Figure 2b, 2c and 5a.

Version 2:

Reviewer comments:

Reviewer #6

(Remarks to the Author)

The authors have seriously addressed my comments and revised the manuscript. I can see clearly the authors' positive attitude and effort to improve the manuscript. However, based on my personal expertise in the field, my own judgement on the novelty and scientific contribution of the work, as well as the potential impact of the work, I insist that the manuscript does not meet the publication criteria of the journal. Nevertheless, I respect other reviewers' opinion and the decision of the editorial office, if they think the manuscript deserves publication.

Reviewer #1 (Remarks to the Author):

The following major remarks require explanation before the decision is made:

- Abstract, line: “...its potential application as a simple and low-cost analytical device that can be operated **without the need of any type of energy**. The device showed high reproducibility and satisfactory detection limits...”

In this case, electrical energy was needed to pump the flow (e.g., by a Syringe pump). I am curious to know by how the flow is produced in the absence of an energy flow that would produce streaming potential. The claims of "low cost" and "no need of any energy" are what make the entire work noble. The liquid itself needed some energy to flow.

We thank the reviewer for the comment.

We recognize that, in an effort to emphasize the novelty of our work, we have exaggerated the concept of “redox reactions driven without electrical power”, and that our device cannot be operated in the absence of some type of energy. We have become aware of this mistake and, hence, considering the reviewer’s suggestion and that we also have to comply with the formatting requirements for this journal (i.e., decreasing the number of words in the abstract), the target sentence in the abstract has been revised as follows:

“The research and implementation of portable and low-cost analytical devices that possess high reproducibility and ease of operation is still a challenging task, and a growing field of importance, within the analytical research. Herein, we report the concept, design and optimization of a microfluidic device based on electrochemiluminescence (ECL) detection that can be operated **without electricity** for analytical purposes. The device functions exploiting the concept of streaming potential-driven bipolar electrochemistry, where a potential difference, generated from the flow of an electrolyte through a microchannel under the influence of a pressure gradient, is the driving force for redox reactions. To our purpose, we have employed such device to drive the ECL reaction of an organic chromophore deposited onto the electrode surface by simply flowing an electrolytic solution containing a coreactant into the microfluidic system, and we have successively applied such device for the detection of amines in water. Our device shows high reproducibility and satisfactory detection limits for tri-n-propylamine,

demonstrating a new, and up to now unexplored, concept of energy saving microfluidic systems with integrated ECL detection.”

- The detection of amine in the water is the foundation for the entire analysis. Nevertheless, the literature section fails to clarify why it came to be identified. I propose that the harmful effects of amines in water on the environment and human health be included. Please read the following literature: Treatment of hazardous organic amine wastewater and simultaneous electricity generation using photocatalytic fuel cell based on TiO₂/WO₃ photoanode and Cu nanowires cathode, Chemosphere, Volume 289, February 2022, 133119.

We thank the reviewer for the comment and for suggesting us this literature, we have read it with very much interest.

We have included the harmful effects of aliphatic and aromatic amines as suggested by the reviewer in the Results session under the subsection “Detection of amines by such electricity-less ECL system”, as follows:

“Based on all the above-reported results, we tested our device for the detection of amines in water-based solutions without the use of electricity. Our interest for amines was justified not only for their pivotal role as coreactants for ECL generation, but also because of their relevance as polluting agents for the environment and dangerousness for human health. Indeed, both aliphatic and aromatic amines are widely used in many industries and, as a result of these applications, they are also dispersed in the atmosphere, water and soil, thereby creating a prospect for human exposure.^{25,43} They possess carcinogenic and mutagenic potential and can cause acute toxicity, including skin sensitization and eye irritation, in workers who are occupationally exposed.⁴⁴ If released in the aquatic environment, they can have toxic effects on aquatic ecosystems and can act as precursors for the formation of several potentially toxic species, including nitrosamines and nitramines.⁴⁵ Hence, their sensitive detection is a matter of primary importance.”

We have also included some additional references relevant to the harmful effects of

amines for the environment and human health, as refs. 25 and 43-45:

(25) Fekete, A., Malik, A. K., Kumar, A. & Schmitt-Kopplin P. Amines in the environment. *Crit. Rev. Anal. Chem.* **40**, 102–121 (2010).

(43) Anastacio Ferraz, E. R., Rodrigues de Oliveira, G. A. & Palma de Oliveira, D. The Impact of Aromatic Amines on the Environment: Risks and Damages. *Front. Biosci. (Elite Ed)* **4**, 914–923 (2012).

(44) Gheni, S. A., Ali, M. M., Ta, G. C., Harbin, H. J. & Awad, S. A. Toxicity, Hazards, and Safe Handling of Primary Aromatic Amines. *ACS Chem. Health Saf.* **31**, 8–21 (2024).

(45) Poste, A. E., Grung, M. & Wright, R. F. Amines and Amine-related Compounds in Surface Waters: a Review of Sources, Concentrations and Aquatic Toxicity. *Sci. Total Environ.* **481**, 274–279 (2014).

• In my opinion, paper-based detection is quite inexpensive and easy to use. Kindly review the following piece of work: Electrochemiluminescence Detection in Paper-Based and Other Inexpensive Microfluidic Devices, *ChemElectroChem* 2017, 4, 1594., <https://doi.org/10.1002/celec.201700426>

We thank the reviewer for pointing out to our attention such interesting minireview. This paper has already been included in the first version of our manuscript as reference n. 17, and it has been cited in the introduction part.

How does the author justify, in this context, that the current flow energy-requiring detection device is "simple," "low cost," and "required no-energy! (see abstract)"? Anyone can claim that the existing paper-based devices is easy to use, inexpensive, and capable of self-capillary driven flow.

We have already recognized our mistake regarding the statement “without the need of any type of energy” that was originally mentioned in the first version of the abstract and we have provided the suitable correction to it as reported in the first reply point of this revision.

In the context of our manuscript, paper-based ECL devices have been mentioned as an

example of inexpensive and portable instruments where ECL detection has been coupled with microfluidics, and for which our prototype can be considered as an additional example of this class of devices. In our case, the “low cost” statement is justified by the fact that our device can be operated without the use of a potentiostat (even if ECL is a process induced ELECTROCHEMICALLY), and the detection analysis (representing a possible application of such device) has been performed in fully aqueous solutions (both distilled and tap water) without the addition of supporting electrolytes, an evidence that can also be considered as a “low cost”, since no additional chemicals are required. In addition, the device can also be operated by hands using a syringe to flow the electrolyte, realizing a real electricity-less system for analytical applications, contributing to a further decrease of the setup costs, while enhancing simplicity of operation and handling of the device. Furthermore, we would also like to point out that it is not our intention to compare pros and cons of our device and paper-based devices, since the underlying theories behind these systems are different, and neither we want to judge which technology is more effective, more practical or more simple to realize. Indeed, our aim is to report such new concept of electricity-less system as it represents an advancement in the fields of analytical and physical-related chemistry and microfluidics.

Also, in the paradigm paper-based detection, optimization of lateral flow assay is important to increase the specificity and efficacy. The authors may look into the following article recently published in this context and can fortify the literatures cited in the introduction part.

Solute imbibition in paper strip: Pore-scale insights into the concentration-dependent permeability, *Physics of Fluids* 35 (12), 2023., <https://doi.org/10.1063/5.0177100>

We thank the reviewer for pointing out to our attention such interesting paper. We understand that the wicking process promoted by capillary action in microfluidic paper is a critical step to control liquid flow in paper-based devices.

However, our prototype represents a different case compared to paper-based devices. Indeed, in our microfluidic system, the fluid movement and charge transport in the microchannel, governed by convective forces under the action of a pressure gradient, are intrinsically coupled with electrokinetic phenomena, promoting the generation of the

streaming effect. Although capillary action may be present in our system during the stasis of the device, for example, inside the porous structure of the resin monolith that composes the filler material of the microchannel, its contribution is not determinant for the functioning of the device. Therefore, we are convinced that the cited literature pertaining streaming effects and electrokinetic phenomena are sufficient for the reader's knowledge.

- As of right now, more technical details regarding the detection don't seem particularly noteworthy because this technique has already been established in the literature: A Wireless Electrochemiluminescence Detector Applied to Direct and Indirect Detection for Electrophoresis on a Microfabricated Glass Device, *Anal. Chem.* 2001, 73, 14, 3282–3288; *Chem. Rev.* 2004, 104, 6, 3003–3036: March 20, 2004, <https://doi.org/10.1021/cr020373d>

We are fully aware of the solidity of the ECL detection in both the academic research and the industrial market. In this regard, we kindly invite the reviewer to refer to refs. 1-4 in our manuscript. We are also aware that the first example on the coupling between bipolar electrochemistry and ECL detection has been reported more than twenty years ago, and that nowadays is an established methodology for analytical applications in a wireless mode, because ECL emission is often used as a reporting methodology of redox reactions occurring on bipolar electrodes. Indeed, the work by Manz et al. referenced by the reviewer (“A wireless electrochemiluminescence detector applied to direct and indirect detection for electrophoresis on a microfabricated glass device”, *Anal. Chem.* 2001, 73(14), 3282-3288) represents, together with the seminal paper by Crooks et al. (“Electrochemical sensing in microfluidic systems using electrogenerated chemiluminescence as a photonic reporter of redox reactions”, *J. Am. Chem. Soc.* 2002, 124(44), 13265-13270), the first report of bipolar electrochemiluminescence. Although in our work the ECL detection itself is common to these studies, it is the novel approach to trigger the ECL emission, namely a **streaming potential** induced by a pressure difference in a bipolar electrochemistry device, and not an **externally applied electric field** typical of the common bipolar systems, that truly represents the novelty of our study and, therefore, we are strongly convinced that it is worthy to be reported, as our is the

first demonstration of this concept.

Regarding the second reference mentioned by the reviewer (“Electrochemiluminescence (ECL)”, Chem. Rev. 2004, 104(6), 3003-3036), it has already been included in the reference list of this work as ref. 7, and, together with ref. 6 (“Electrogenerated chemiluminescence and its biorelated applications”, Chem. Rev. 2008, 108(7), 2506-2553), it offers to the reader a comprehensive understanding of the ECL detection, from the basic principles to its applicability as highly sensitive electroanalytical technique.

Reviewer #2 (Remarks to the Author):

The manuscript presents ECL measurements using a (split) bipolar electrode. The experiments are described in detail, including the rationalization of several design choices. I however believe that the motivations behind this work are overstated, in particular:

1. The authors emphasize that a major benefit of the device is that it can be operated without external electrical power source, for example for use “in regions that do not have ready access to electricity”. I believe that this aspect is severely overstated since a simple battery would be sufficient to drive the ECL experiment. This is much simpler and practical than employing an external pump to induce a streaming potential through a microfluidic cell. I understand that the basic phenomenon can be produced by hand with a syringe, but presumably the flow rate will not be accurate enough for analytical applications in that case.

We thank the reviewer for the remark. We recognize that the first version of our manuscript’s introduction part may appear exaggerated, since, as suggested by the reviewer, there are simpler methods to achieve ECL emission. We have revised the second part of the introduction section considering a more realistic scenario, and by focusing on the combination between ECL detection and microfluidics that is still a growing area of research, especially in the context of sensing and analytical chemistry.

We have revised that part of the introduction part as follows:

“In this regard, the few hardware requirements of ECL is of great advantage for its integration into miniaturized systems specifically designed for sensing purposes, with the aim to achieve low limits of detection while reducing sample volumes. Indeed, some examples of such integrated systems have been recently reported. For instance, smartphones with built-in camera technology having excellent sensing capabilities for analytical purposes can be used for ECL detection, providing real time analysis through the fusion of widespread consumer hardware with networked technologies.¹⁶ Similarly, microfluidic paper-based ECL devices combine minimal reagent consumption with a cheap and disposable platform for fast detection in a single portable instrument.¹⁷ In particular, the integration of ECL detection into microfluidic systems has been reported

as an effective method for carrying out highly sensitive analysis in low sample volume, and by employing portable devices with high reproducibility and ease of operation.¹⁸ Therefore, the continuous search of innovative approaches to combine microfluidics with ECL detection is still a challenging goal to address towards the implementation of analytical strategies aiming to cost reduction and minimal human handling.”

The revised part includes the following three references:

(16) Doeven, E. H., Barbante, G. J., Harsant, A. J., Donnelly, P. S., Connell, T. U., Hogan, C. F. & Francis, P. S. Mobile Phone-Based Electrochemiluminescence Sensing Exploiting the 'USB On-The-Go' Protocol. *Sens. Actuators, B* **216**, 608–613 (2015).

(17) Gross, E. M., Durant, H. E., Hipp, K. N. & Lai, R. Y. Electrochemiluminescence Detection in Paper-Based and Other Inexpensive Microfluidic Devices. *ChemElectroChem* **4**, 1594–1603 (2017).

(18) Kirschbaum, S. E. K. & Baeumner, A. J. A Review of Electrochemiluminescence (ECL) In and For Microfluidic Analytical Devices. *Anal. Bioanal. Chem.* **407**, 3911–3926 (2015).

In addition, we have also revised the statement related to the “regions that do not have ready access to electricity” in the Discussion section with the more general term “in areas with limited access to electricity”, as follows:

“... We believe that our prototype represents a novel class of low-cost, portable and electricity-less electrochemical devices that can be employed for sensing applications in emergency situations or in areas with limited access to electricity.”

2. Furthermore, direct electrical actuation would be far more energy efficient (another purported advantage of the approach) as it would eliminate the very significant losses from viscous drag in the fluid. Based on the numbers in the manuscript, the mechanical power needed to drive the device is ~20 mW, about 4 orders of magnitude higher than the power consumed by the faradaic process (order 1 uW). This is highly inefficient.

The energy conversion from a streaming potential source is an important point as arisen from the reviewer and we recognize it here.

We are aware of the low efficiency of the approach, as we also had previously calculated the energy requirements for operating the device, as reported in the following procedure. In order to estimate the energy requirement of the device, we calculated the fluid power generated for a given ΔP and flow rate:

$$\text{Fluid power (in Watt)} = \Delta P \times \text{volumetric flow rate}$$

The minimum value of ΔP required to generate streaming potential with the phenolic resin monolith no. 1 as the filling material is 0.5 MPa as shown in Supplementary Figure 3, hence:

$$\Delta P = 0.5 \text{ MPa} = 0.5 \times 10^6 \text{ kg/ms}^2$$

$$\text{With pressure unit: Pa} = \text{N/m}^2 = \text{kg/ms}^2$$

The value of the flow rate chosen for the optimal device operation is 0.3 mL/min, hence:

$$\text{Volumetric flow rate} = 0.3 \text{ mL/min} = 4.8 \times 10^{-9} \text{ m}^3/\text{s}$$

$$\text{With rate unit: mL/min} = 10^{-6} \text{ m}^3/60 \text{ s} = 1.6 \times 10^{-8} \text{ m}^3/\text{s}$$

Therefore, the fluid power needed to drive the device is:

$$\text{Fluid power} = (0.5 \times 10^6) \times (4.8 \times 10^{-9}) \text{ kgm}^2/\text{s}^3 (= \text{J/s} = \text{W}) = 2.4 \times 10^{-3} \text{ W} = \mathbf{2.4 \text{ mW}}$$

We compared this value with the power consumed by redox processes developed at the split BPE system, according to the following equation:

$$\mathbf{1 \text{ W} = 1 \text{ V} \times \text{A}}$$

and

according to Supplementary Figure 5, the value of the threshold potential necessary to start the ECL process is 1 V;

the current measured with the split BPE system using the same conditions of Supplementary Figure 5 at 0.3 mL/min is $\sim 1.2 \mu\text{A}$ (according to Entry 6 in Supplementary Table 2), hence:

$$\text{Power (redox process)} = 1 \text{ V} \times 1.2 \times 10^{-6} \text{ A} = 1.2 \times 10^{-6} \text{ W} = \mathbf{1.2 \mu\text{W}}$$

From our evaluation, we observe that the power needed to drive the device is around one order of magnitude smaller than the value calculated from the reviewer. However, the

trend observed is the same. Because, our aim is to report the innovative concept of microfluidic system with integrated ECL detection operated by streaming potential for analytical applications, we have not studied or optimized the device to improve its efficiency, as this is not a system to convert mechanical energy (the fluid flow) into electrical energy (energy from redox process). In addition, in the discussion we have not mentioned the energy requirements of the system in terms of “another purported advantage” of the approach.

3. Wider applications are mentioned, for example in the realm of diagnostics. However, the experiments presented here are performed at very low electrolyte concentrations, ~100x smaller than physiological salt concentrations. Since the streaming potential is suppressed at high ionic strength, it is far from clear that the approach could be deployed to this broader class of systems. Neutral pH could also lead to a decrease of the zeta potential of the channel medium, further decreasing the streaming potential.

We thank the reviewer for the remark.

We recognize the limitation of our method for applications regarding diagnostics and healthcare monitoring. As discussed by the reviewer, the flow of fluids with high ionic concentrations may result in a decrease of streaming potential and, therefore, in a decrease of the ECL detection.

Considering the reviewer’s remark, we have removed from our manuscript all statements regarding the potential application of the method for diagnostics and healthcare monitoring, while we have included only the analytical detection for the environment, namely detection of amines in water. This revision particularly interests the introduction and Discussion sections of our manuscript.

4. Although the authors claim to have performed “an extensive investigation of all the experimental parameters in play”, there is an important omission. During operation, the two electrodes are short-circuited via an ammeter. Were the electrodes close to non-polarizable, no significant streaming potential could develop. Instead, a streaming current

would exist that would loop back through the ammeter. The fact that a streaming potential develops thus depends on the size and nature of the electrodes, such that some degree of polarization is possible. This is however not addressed in the manuscript, making it practically impossible to generalize the experimental results shown here to a more general situation.

We thank the reviewer for the comment and for highlighting this important point regarding the effect of the electrode material on the performances of the device. We apologize for omitting this important investigation. Therefore, we have performed new experiments to clarify this point.

Unfortunately, due to our cell specifications, it was not possible to change the diameter of the wires (electrodes), so that we had to keep this parameter constant at 0.6 mm. However, changing the type of material was possible, and including platinum (Pt), we have additionally investigated gold (Au), silver (Ag) and copper (Cu). We have not observed substantial differences in the E_{str} values among the different metals, while ECL emission was strongly influenced by the choice of the electrode material.

We have included these new results in the supporting information, as follows:

“13. Impact of the electrode material on the ECL performances of the device

In order to fully evaluate the ECL performances of the device, we also investigated the effect of the electrode (anode) material of the split BPE system. To this purpose, we have replaced the Pt wire acting as the anode with different metals, such as gold (Au), silver (Ag) and copper (Cu), and by dipping each wire in a 20 mg/mL of BTD-TPA solution and then drying them at the air. All wires used had a constant diameter of 0.6 mm. Phenolic resin monolith no. 2 was employed as the filling material. An electrolyte-free solution composed of MeCN/H₂O (3:1 v/v) containing 1 mM TPrA was flown at 0.3 mL/min. The results are summarized in Supplementary Table 7 and Supplementary Fig. 10.

Supplementary Table 7. Effect of the electrode material on the ECL generation when flowing an electrolyte-free solution composed of MeCN/H₂O (3:1 v/v) containing 1 mM TPrA at the flow rate of 0.3 mL/min. All wires had a constant diameter. Cathode material

was Pt ($\phi = 0.6$ mm) in all cases.

Entry	Material	ΔP [MPa]	E_{str} [V]	i [μA]	Maximum ECL intensity [a. u.]	Integrated ECL intensity [a. u.]
1	Pt	2.1	5.68	0.69	0.30	58.8
2	Au	2.1	5.78	0.69	0.11	25.9
3	Ag	2.1	5.83	0.69	0.00	0.00
4	Cu	2.1	5.88	0.71	0.00	0.00

Supplementary Figure 10. ECL intensity of the BT-D-TPA film in air during the pumping of the solution for 360 s with the streaming potential device using different electrode materials as the anode constituting the split BPE system: Pt (light blue line), Au (orange line), Ag (dark blue line) and Cu (green line). Conditions: 1 mM TPrA in an electrolyte-free MeCN/H₂O (3:1 v/v) solution flown at the flow rate of 0.3 mL/min. Phenolic resin monolith no. 2 was employed as the filling material. PMT bias = 1000 V.”

To discuss these results, we have added the following paragraph in the subsection “Generation of ECL powered by streaming potential” of the manuscript:

“It is well known that the nature of the electrode, such as the material and its surface state, strongly influences the ECL emission process.³⁹ Generally, carbonaceous materials are good candidates for ECL applications because of their low cost, disposability and for the efficient TPrA oxidation.⁴⁰ However, metallic electrodes are usually preferred because of

their stability and fast kinetics of electron transfer reactions. Therefore, we next investigated the effect of the electrode material on the ECL performances of the device, including the generation of streaming potential. To this aim, we replaced the Pt anode of the split BPE system with different metals, such as gold (Au), silver (Ag) and copper (Cu). As shown in Supplementary Fig. 10, Pt showed the highest ECL intensity but with a pronounced decay, while Au showed a relatively stable, but lower ECL emission. Generally, the ECL performances of the Au material are superior to those of Pt, because passivation of the electrode surface, that is due to the generation of an oxide layer that negatively impact TPrA oxidation, occur at more positive potentials compared to Pt.^{39,41} However, it is accepted that on Pt electrode ECL is mainly generated via the catalytic pathway,^{34,42} which does not involve direct TPrA oxidation but rather its homogeneous chemical oxidation by the oxidized chromophore, a fact that might explain the highest ECL intensity generated in the present case. ECL emission could not be detected using the less noble metals Ag and Cu, although the BTD-TPA films were formed on the electrodes and no apparent metal dissolution was observed. Nevertheless, the use of Cu generated the highest current flowing through the split BPE system (Entry 4 in Supplementary Table 7), an evidence that might be ascribed to copper oxidation that generally occurs at relatively mild potentials. In addition, the different nature of the electrodes did not significantly impact the magnitude of E_{str} (Supplementary Table 7), since 0.2 V is the difference between the highest and the lowest E_{str} values generated from the use of Cu and Pt, respectively.”

Please, kindly refer to the listed references in the manuscript.

Some additional points to improve the manuscript include:

5. The statement that the device “can be operated without the need of any type of energy” (abstract, page 2) is clearly incorrect as mechanical energy powers the experiment. Statements elsewhere in the manuscript are more carefully worded, and it would be preferable to be more precise in the abstract as well.

We thank the reviewer for the comment.

Considering the reviewer's suggestion, the target sentence in the abstract has been deleted and the abstract has been revised as follows:

“The research and implementation of portable and low-cost analytical devices that possess high reproducibility and ease of operation is still a challenging task, and a growing field of importance, within the analytical research. Herein, we report the concept, design and optimization of a microfluidic device based on electrochemiluminescence (ECL) detection that can be operated **without electricity** for analytical purposes. The device functions exploiting the concept of streaming potential-driven bipolar electrochemistry, where a potential difference, generated from the flow of an electrolyte through a microchannel under the influence of a pressure gradient, is the driving force for redox reactions. To our purpose, we have employed such device to drive the ECL reaction of an organic chromophore deposited onto the electrode surface by simply flowing an electrolytic solution containing a coreactant into the microfluidic system, and we have successively applied such device for the detection of amines in water. Our device shows high reproducibility and satisfactory detection limits for tri-n-propylamine, demonstrating a new, and up to now unexplored, concept of energy saving microfluidic systems with integrated ECL detection.”

6. It is unclear what the pressure drop is in the inlet and outlet tubing, and hence the statement “the discharge pressure displayed by the pump is the internal pressure of the upstream chamber” could be further justified.

We thank the reviewer for the comment. In our study, we have not used a pressure gauge to measure the pressure inside the chambers of the microfluidic device. Hence, we have revised the target sentence (located in the Supporting Information document, page S5) as follows: “**The value of the pressure displayed by the pump reflects the discharge pressure for the fluid. In our device,** this pressure is gradually reduced across the microchannel to **reach** the atmospheric pressure in the downstream chamber, implying that ΔP can be monitored and tuned with the feeding pump by changing the flow rate.”

7. The term “ECL integral area” is not very intuitive, this terminology could be adjusted for clarity.

Following the reviewer’s suggestion, we have revised all terms reporting “ECL integral area” with the more commonly used “integrated ECL intensity”. This revision interests all terms mentioned in the manuscript and Supplementary Information, including all Supplementary Tables and Figure 3d and Supplementary Figure 6g.

8. The text is lucidly written and easy to follow. There are however a few language mistakes, for example “an extensive investigations” (page 2), “whit” (page 7), and “did not significantly changed” (page 12).

We thank the reviewer to point out to our attention such language mistakes. According to the reviewer’s suggestion, we have revised only one point remaining from the first version as follows:

- page 8 in this version, “whit” → “with”
- the sentence “did not significantly changed” related to the discussion reported in page 14 of this version has been deleted due to the inclusion of the new Supplementary Fig. 14
- the sentence “an extensive investigations” (abstract, page 2) mentioned in the previous version of the abstract has been deleted in the current version due to a decrease in the number of words to respect the formatting requirements for this journal.

Reviewer #3 (Remarks to the Author):

In this article, the authors report a fluidic device that accomplishes electrochemiluminescence (ECL) detection for amines in a flowing solution, harnessing the concept of streaming potential-driven bipolar electrochemistry. Pressure driven flow through a microchannel filled with a porous material generates a streaming potential, which in turn, drives ECL reactions at the anode of a bipolar electrode (BPE) that spans the channel. Experimental parameters including properties of the filler material, flow rate, electrolyte composition, chromophore's loading onto the electrode, and the type of co-reactant are investigated. The device achieves a limit of detection (LOD) of 0.025 mM for tripropylamine (TPrA) in MeCN/H₂O (3:1 v/v) solution without requiring supporting electrolytes or buffered solutions. The ECL intensities in fully aqueous systems are low compared to the results obtained using the mixed solvent system.

This work is important because it addresses challenges to the practical application of pressure-driven bipolar electrochemistry including 1) incorporation of ECL reagents (luminophore) into a solid phase on the BPE anode, 2) a need for larger streaming potentials, and 3) removal of the need for electrolytes or buffered solutions. The device developed and characterized in this study operates without external power, is low-cost, and is portable — ideal for sensing applications in emergencies or regions with limited access to electricity.

This manuscript is well-structured, and the figure captions and legends are very clear. However, certain sections of the article would benefit from additional discussion to better explain the experimental results and fundamental mechanisms. Further, the potential for practical application is not argued strongly enough.

We thank the reviewer for the positive comments to our manuscript and for his/her contribution to further improve our work.

Major Comments:

1. The data presented here is primarily collected in 3:1 MeCN:H₂O as the solvent to

enhance the ECL signal. While signal was obtained in distilled water and tap water, even 1 mM TPrA (model analyte) produced a weak signal. Under what conditions, theoretically, could the reported system be useful for sensing amines at environmentally or clinically relevant concentrations?

We thank the reviewer for the comment.

In our revised manuscript, after additional experimental work and consideration of all the experimental data, we have decided to remove any reference to the potential application of our method for clinical diagnostics and healthcare monitoring, since solutions with high ionic conductivity may suppress the generation of streaming potential and, hence, hinder the ECL detection of the device. Therefore, the method seems more suitable for applications such as environmental detection and water control. Our reported concept has been primarily developed in a mixed solution composed of MeCN:H₂O, and then applied to fully aqueous solution systems with relatively good performances of the devices. To the best of our knowledge, our study represents the first example of generation of ECL in a fully aqueous system without the use of any supporting electrolytes or buffered solutions.

2. Page 8, line 121. The authors indicate that the current flowing through the BPE is the sum of the faradaic current and the streaming current. However, in this system the streaming current and the faradaic current flow in series. The streaming current is the sum of the faradaic current and the charging current (non-faradaic). Further, on page 8, line 210, the authors indicate that it is difficult to estimate the current efficiency (percent of current being used to drive ECL) because of the mixed contribution of faradaic and streaming current in this system. The reviewer's understanding is that it is faradaic and charging current that are additive (not streaming current), and therefore, the current efficiency can be obtained following the decay of charging current. Further, if needed, the authors can operate this same device in an electrolytic mode (e.g. amperometric or voltammetric) to evaluate efficiency.

We thank the reviewer for this interesting comment and for the kind suggestion regarding evaluation of the current efficiency in our system.

Our apparatus for current measurements as shown in Supplementary Fig. 1 exploits an

ammeter that is connected between the two Pt wires (electrodes) outside the fluidic space. This system corresponds to a split BPE and implies that the current data obtained and reported in this work (i values) are average values obtained during the device operation. Therefore, at the present time, it is not possible to obtain the classical current/time profiles and the behavior of the current during the operation of the device cannot be further elucidated.

Nevertheless, considering the reviewer's remark, we have corrected this discussion in the manuscript as follows:

“In our streaming potential device, the non-faradaic current (streaming current^{35,36}) is counterbalanced by the ohmic current and the bipolar faradaic current,³⁷ the latter derived from the redox processes developed at the split BPE system. In other words, the streaming current can carry part of all current generated electrochemically. In addition, faradaic currents generated at electrodes embedded in fluidic channels can also generate electroosmotic flows in a manner that, coupled with convection, can generate effects complementary to streaming current and potential.³⁵ Hence, at this stage of our investigation, it is rather difficult to quantify the magnitude of the faradaic current flowing through the split BPE system.”

Please, kindly refer to the listed references in the manuscript.

In addition, considering that the faradaic current is not exclusively converted into photons (i.e., the ECL process), and that in bipolar systems it is not possible to know exactly the potential at the electrodes (at the anode in our case), we cannot exclude that additional processes, like the water oxidation reaction, can occur concomitantly to the ECL reaction, while contributing to the magnitude of the total faradaic current. Therefore, and also considering the area of application of our device, we believe this is not a parameter of paramount importance for the development of the device.

3. Page 13, line 321. The authors indicate that the low sensitivity of this system may be caused by lack of a buffer (to control deprotonation of the co-reactant). The authors may

consider other mechanisms for the low sensitivity. It is probable that the TPrA that is oxidized at the BPE anode is being swept away by the flowing solution prior to subsequent reaction with the luminophore. Potential evidence of this effect would be a decay in the ECL intensity at very high flow rates. If this is the case, then the signal would be enhanced by positioning the electrodes in stagnant fluid outside of the direct line of fluid flow. Along the same line of thought, differences in ECL intensity among the three co-reactants tested may have an underlying kinetic mechanism. Could the authors comment on the potential for TPrA radical to be swept away prior to reacting with the luminophore/coating?

We thank the reviewer for the question and for highlighting this important point concerning the intimate correlation between ECL (streaming potential) and flow rate.

We have studied the effect of the flow rate on the ECL intensity and, as reported in Figure 3d, we have found that the intensity of light emission increased by increasing the flow rate. The ECL intensity profiles recorded in these conditions, as reported in the Supplementary Figure 4, showed that the ECL emission was not quenched at high flow rates even after several minutes that the device was in operation (please, see the ECL profile of Supplementary Figure 4f at 300 sec). These results indicate that fresh TPrA (coreactant) is continuously provided at the anode for the ECL reaction. This evidence is also corroborated from ECL measurements performed in different solvents, such as the mixed MeCN/H₂O solution with or without electrolyte (Figure 3c) and in the fully aqueous solutions (with lower intensities, Figure 5c), that all showed continuous light emission without quenching, suggesting that depletion of TPrA is not occurring by increasing the solution flow rate or by changing the type of solvent.

We have found this interesting paper recently published: “Electrochemiluminescence in microfluidic channels: influence of mass transport on the tris(2,2'-bipyridyl)ruthenium(II)/tripropylamine system at semitransparent electrodes” (Anal. Chem. 2024, 96, 14650-14659, DOI: 10.1021/acs.analchem.4c03344). Here, it is reported that under flow conditions when mass transport is controlled by diffusion (low flow rates), TPrA is depleted at the electrode surface resulting in a quench of the light emission close to the electrode (comparison of concentration profiles and ECL mappings at different flow rates as reported in Figures 5 and 6 of the above-mentioned paper). When convective

regime prevails instead (high flow rates), the solution flow continuously delivers fresh TPrA all over the electrode surface, resulting in ECL to occur on its entire surface.

Considering that in our study we have employed high flow rates for the optimal device operation and that mass transport is likely controlled by convection in a such a case, we recognize that the experimental evidence reported in the above mentioned manuscript belongs to our case. Therefore, we have added this consideration in the manuscript, in the section “Generation of ECL powered by streaming potential” when discussing Figure 3d, as follows:

“The highest ECL intensity and the largest integrated ECL intensity were obtained by increasing the flow rate up to 0.3 mL/min (Fig. 3d). The ΔP had to be controlled within 5 MPa because larger pressures in the upstream chamber would cause the breakage of the quartz cell. Therefore, the maximum flow rate was limited to 0.3 mL/min. This condition is also appropriate for the analytical application of the device, because mass transport is controlled by convection regime that provides constant flow of fresh TPrA over the electrode surface assuring ECL generation over time.³⁴”

With ref. 34:

(34) Ma, Y., Sella, C. & Thouin, L. Electrochemiluminescence in microfluidic channels: influence of mass transport on the tris(2,2'-bipyridyl)ruthenium(II)/tripropylamine system at semitransparent electrodes. *Anal. Chem.* **96**, 14650–14659 (2024).

4. Page 14, line 358. Since there are low or no supporting electrolytes in the fully aqueous systems, the solution conductivity is very low. Is there a possibility that the low ECL signal is limited by not having enough charge carriers in the solution? Under certain conditions, is it possible that the streaming current limits the current through the entire circuit (including BPE current)?

We thank the reviewer for arising this interesting point regarding the use of low concentrated supporting electrolytes (and without) to operate the device. To the best of our knowledge, we are the first to report the generation of an ECL process in fully aqueous solutions without electrolytes (unbuffered solutions) using bipolar electrochemistry, so there are no other references in literature to compare our results.

In our previous work (“Electropolymerization without an electric power supply”, *Commun. Chem.* **2022**, 5, 66), we have found that the use of very low concentrated electrolytes (and low solution conductivities) is beneficial for the generation of streaming potential and that streaming potential values suitable for redox reactions can also be achieved without the addition of the supporting electrolyte in the flowing solution. Therefore, there should not be any limitation imposed by the streaming current for operating the device, while we believe this point pertains mainly to the ECL mechanism. We have found in literature that decreasing the ionic strength of the electrolyte improves the ECL efficiency of the ruthenium/TPrA system, because faster TPrA deprotonation occurs (from TPrA radical cation to generate TPrA radical, both responsible of the ECL process) (“Effect of ionic strength on the electrochemiluminescence generation by Tris(2,2'-bipyridyl)ruthenium(II)/Tri-n-propylamine”, *Chem. Res. Chin. Univ.* **2022**, 38, 816–822). We also believe that this evidence belongs to our system.

Considering the reviewer’s comment, we have added the following considerations in the discussion of our results, as follows:

(a) Section “Generation of ECL powered by streaming potential”, when discussing the different ECL intensities using 0.5 mM supporting electrolyte and without it: “This effect may be ascribed to the generated cationic species of BTD-TPA and TPrA that, in solution without electrolyte, are naked, i.e., such radical cations are not stabilized by the anions derived from the supporting electrolyte and may be active for the ECL reaction. In particular, the low ionic strength of the electrolyte increases the ECL efficiency of TPrA, due to the increase of the deprotonation rate of the TPrA radical cation ($\text{TPrA}^{\bullet+}$) to generate the TPrA radical (TPrA^{\bullet}) species,³⁸ that are both fundamental in the ECL generation process.”.

With ref. 38: (38) Hu, S., Wang, Y. & Su, B. Effect of ionic strength on the electrochemiluminescence generation by Tris(2,2'-bipyridyl)ruthenium(II)/Tri-n-propylamine. *Chem. Res. Chin. Univ.* **38**, 816–822 (2022).

(b) Section “Detection of amines by such electricity-less ECL system”, when discussing the observed ECL trend of the three different coreactants: “Hence, this evidence might explain the progressive decrease of the ECL intensity by increasing the number of the electron-withdrawing groups on the amine, giving the order TPrA>DBAE>TEOA for

our streaming potential device. Nevertheless, it is important to underline that in our system the substitution effect may not be the determining factor for explaining the observed trend, since the experimental conditions used to operate the device are far from the common ones employed in ECL analysis. Even though such conditions are ideal for the buildup of E_{str} values sufficiently large to drive redox processes on the Pt electrodes, they also undoubtedly affect the efficiency of the coreactant. For instance, it has been reported that low concentrations of DBAE produce higher ECL intensities compared to TPrA in the same concentration range.⁵⁵ Indeed, when the streaming potential device was operated at low flow rates up to 0.20 mL/min, 1 mM DBAE generated higher ECL intensity compared to 1 mM TPrA (Supplementary Figs. 16(a)-(d) and integrated ECL intensity values in Supplementary Table 11). However, at high flow rates, the trend reversed and TPrA generated the highest ECL emission. Since the radical intermediate of DBAE is much more unstable than that of TPrA,⁵⁶ the high flow rate presumably decreases the efficiency of this coreactant in our system.”

With refs. 55: (55) Liu, X., Shi, L., Niu, W., Li, H. & Xu, G. Environmentally friendly and highly sensitive ruthenium(II) tris(2,2'-bipyridyl) electrochemiluminescent system using 2-(dibutylamino)ethanol as co-reactant. *Angew. Chem. Int. Ed.* **46**, 421–424 (2007), and 56: (54) Sentic, M., Milutinovic, M., Kanoufi, F., Manojlovic, D., Arbault, S. & Sojic, N. Mapping electrogenerated chemiluminescence reactivity in space: mechanistic insight into model systems used in immunoassays. *Chem. Sci.* **5**, 2568–2572 (2014).

Comments:

1. Page 2, line 40. The statement saying that the device can be operated without the need of any type of energy is imprecise.

We have deleted this sentence in the abstract and modified some statements to comply with the formatting requirements for this journal (i.e., decreasing the number of words in the abstract). Considering the reviewer’s comment, the concept of energy in the new version of the abstract has been revised as follows:

“Herein, we report the concept, design and optimization of a microfluidic device based

on electrochemiluminescence (ECL) detection that can be operated without electricity for analytical purposes.”

2. Page 2. The sensitivity of the device and LODs of the amines are not mentioned in the abstract.

As we have been requested to comply with the formatting requirements for this journal, we have to decrease the number of words in the abstract. Because the sensitivity of the device is not the crucial result of our work, we have not included the LOD for tri-n-propylamine in the abstract. We believe it is fine to discuss it in the “Results session” of our manuscript.

3. Missing details in both the article and the SI on how the area and thickness of the BTD-TPA film deposited on the Pt wire electrode are controlled.

We have investigated the effect of the BTD-TPA loading on the ECL performances of the device and we have reported these data in the Supplementary Information document under the section “Optimization of the amount of BTD-TPA coating onto the anode”. The procedure for preparing the BTD-TPA film on the Pt electrode is reported in the same document under the section “BTD-TPA coating onto the electrode”. To control the loading of the chromophore on the electrode, we simply changed the concentration of the solution used for preparing the film, while the area of the film was limited by the area of the Pt electrode immersed in the solution. In all cases, the wires used had the same diameter (0.6 mm) and the same length, so that the same area of the wire was coated with the BTD-TPA film.

4. S10. What is the difference between the 6 runs shown in Figure S4 (b) and (c)?

Supplementary Figures 5b and 5c of this version report the cyclic voltammetry and the simultaneous ECL emission recorded from a BTD-TPA film using the measurement system as reported in Supplementary Figure 5a. Therefore, Supplementary Figure 5b

reports the current values vs. voltage (current-voltage curves), while Supplementary Figure 5c reports the ECL intensity vs. voltage (ECL intensity-voltage curves). This information is reported in the legend of Supplementary Figures 5b and 5c.

5. Page 7, line 192. Could the authors provide an explanation as to why the ECL signal reaches the maximum intensity immediately and then gradually decays?

According to the reviewer's request, we have provided an explanation for a typical ECL profile in the section indicated as follows:

“The generation of a light emission from the split BPE anode was detected soon after feeding the device with the electrolytic solution, which immediately reached the maximum intensity, followed by a gradual decay (Fig. 3c, orange line, Supplementary Table 2 and Supplementary Fig. 4). Indeed, when E_{str} reaches the suitable value to trigger the ECL reaction, both the chromophore and the coreactant, namely TPrA, are oxidized at the electrode surface. The oxidation ignites the cascade reactions that prompt the step increase of light intensity. In this first step, the TPrA located around the electrode is fully consumed by the oxidation process and, subsequently, fresh TPrA has to move toward the electrode to sustain the light generation. The mass transport process is governed by TPrA diffusion and forced convection induced by the flow of the electrolyte. Nevertheless, a gradual decay of the light emission is concomitantly observed, likely due to the physical or chemical deterioration of the BTD-TPA coating onto the electrode during the oxidation process. On the other hand, the continuous flow of fresh electrolyte can supply enough coreactant to sustain the light emission even for several minutes of device operation. In such a case, the process is only under convection control by the flowing electrolyte, while the diffusion layer thickness of TPrA is constant, giving to the ECL emission a constant profile typical of a convection-controlled process.³⁴”

With ref. 34:

(34) Ma, Y., Sella, C. & Thouin, L. Electrochemiluminescence in microfluidic channels: influence of mass transport on the tris(2,2'-bipyridyl)ruthenium(II)/tripropylamine system at semitransparent electrodes. *Anal. Chem.* **96**, 14650–14659 (2024).

6. Page 9, Figure 3 (c) and (e). These two figures appear to be slightly repetitive; the author can consider deleting Figure 3 (c) and combining the captions for the two figures. Following the reviewer's suggestion, we have combined Figures 3(c) and 3(e) in a single figure, the current Figure 3c. This revision has also been reflected in the legend of Figure 3c and in the text of the manuscript, where the reference to the former Figure 3(e) has been removed or adjusted according to the new Figure 3.

7. Page 9, Figure 3 (e) shows that the ECL signal of the electrolyte-free solution is higher than that of the same solution with added electrolyte. How do the corresponding current signals compare?

The value of the current observed with the ammeter during operation of the device is constant during the ECL reaction. This trend is evident during the on/off operation of the device, where the pulsed flow shows constant peak currents, as reported in Supplementary Figure 13. Hence, the comparison of the current values between the two cases reported in Supplementary Table 5 is reasonable. Even if the current values were very similar, the ECL performance of the device in the electrolyte-free condition was better.

8. Page 10, line 228. This statement needs to be clearer. The low solution conductivity doesn't contribute to onset, but to magnitude.

According to Smoluchowski equation, streaming potential and conductivity are inversely proportional, hence, a decrease in the solution conductivity gives rise to streaming potential. For this reason, the sentence as reported in the manuscript is correct. To support this statement, we have recalled ref. 35 (Kostiuchenko, Z. A., Cui, J. Z. & Lemay, S. G. *Electrochemistry in Micro- and Nanochannels Controlled by Streaming Potentials. J. Phys. Chem. C* **124**, 2656–2663 (2020)), where the same concept is illustrated at the end of the introduction part at page 2657.

9. Page 10, line 249 and line 254. What's the authors' explanation for why a concentration of 1.0 mM TPrA and pH 10 exhibit the best ECL performance?

Low concentrations of TPrA, like 1 mM, are completely soluble in aqueous solutions, but the solubilization of the amine causes an increase of solution pH. In our case, we did not decrease the pH to values close to the neutrality because no additional electrolytes were added in the electrolyte-free MeCN/H₂O (3:1 v/v) solution for not modifying the solution conductivity. Furthermore, even if TPrA is an efficient coreactant for ECL generation, high concentrations of this amine can lead to a quenching of the emitting excited state. For all these reasons, 1 mM TPrA concentration and pH 10 were the optimal conditions for the device that we found from our investigations.

10. S19, Table S7. The solution conditions are the same, why is there a significant difference between the ΔP and E_{str} of entry 1 and entry 2? Both ISO and exposure time are set differently for the still imaging and video, the results should be incomparable.

The two entries of Supplementary Table 8 in this version have the same solution composition, but the flow rate is different between the two cases and, hence, different values of ΔP and E_{str} were generated. We have used the highest flow rate (0.50 mL/min) for recording the movie of an ECL emission generated with the device. Because our camera has different ISO settings for recording movies or still images (lower ISO values for movies compared to the ISO for still images), we have recorded both movies and still images by using the camera highest sensitivity and by adjusting the flow rate of the device for each case. However, in our manuscript we have **NOT** directly compared the ECL intensity of a still image with that of a movie, but we have included these data, as those reported in Supplementary Table 8, Supplementary Figure 11 and Supplementary Movie 1, only for completeness of information.

11. S20, Figure S9 (i). How is this plot generated? Given that the data from the digital camera's still imaging should yield one ECL signal every 30 seconds, the methodology for producing this plot requires clarification.

The graph reported in Supplementary Figure 11i in this version corresponds to a single ECL signal recorded using the PMT and by employing the conditions of entry 1 in Supplementary Table 8 (i.e., a solution composed of 1 mM TPrA in an electrolyte-free MeCN/H₂O (3:1 v/v) flown through the device at 0.30 mL/min). This graph, and the method used to generate it, is the same respect to all the others ECL intensity graphs reported in this study. The graph in Supplementary Figure 11j was recorded with the same conditions of the graph 11i, but using a higher flow rate of 0.50 mL/min as reported in entry 2 of Supplementary Table 8. Indeed, this is also inferred by the shape of the two ECL profiles that are very similar, but the intensity of the emission is higher for the graph in 11j because of the higher flow rate applied. Regarding the (c) to (h) letters reported in 11i, these correspond to the location and duration of the camera acquisition (30 sec) applied to record the still images reported in Supplementary Figures 11c to 11h using the setup shown in 11a. For example, considering the case of the (c) letter in graph 11i, camera acquisition started at the beginning of the solution flow, where the E_{str} is not yet enough to trigger the ECL reaction, and lasted 30 sec, giving a dark image (11c) where ECL emission was not detected. On the contrary, for the case of the (d) letter, camera acquisition started during the light emission and after 30 sec of acquisition resulted in a bright ECL image of the BTd-TPA coating located inside the quartz cell. This procedure has also been applied to the remaining (e) to (h) cases. Indeed, the brightness in 11h is substantially decreased compared to the image in 11d, because the acquisition has been performed after 300 sec of device operation.

Considering the reviewer's remark, we have revised the legend of Supplementary Figure 11 as follows: "Supplementary Figure 11. (a) Setup for the simultaneous ECL imaging acquisition using a digital camera (located on the left side of the quartz cell) and recording of the ECL intensity profile using a PMT (located on the right). (b) Photograph of the quartz cell under natural light showing the BTd-TPA coated electrode. (c–h) Collected ECL images (exposure time 30 sec each) during the flow of the solution using the conditions of entry 1 in Table S7 at (c) 0 sec, (d) 100 sec, (e) 150 sec, (f) 200 sec, (g) 250 sec, and (h) 300 sec of device operation. (i, j) ECL intensity of the BTd-TPA film in air during the pumping of the solution for 360 s with the streaming potential device measured

using the conditions of Table S7, for (i) entry 1 and (j) entry 2. PMT bias = 1000 V. Camera parameters for still imaging: ISO 25600, f/2.8, exposure time 30 sec. Camera parameters for movie: ISO 12800, 30 fps. The (c) to (h) letters indicated in graph (i) correspond to the location and duration of the camera acquisition applied to record the still images reported in panels (c) to (h).”

12. Page 12, line 287-289. How stable is the BTD-TPA coating? How long can it last and is it reusable?

We have checked the stability of the BTD-TPA coating onto the Pt wire by recording 3D laser microscope images of a pristine BTD-TPA film and after the on/off operation of the device repeated for 5 and 15 times as reported in the new Supplementary Figure 14, please kindly see the reply to the next question. As mentioned in the manuscript, a physical and/or chemical deterioration of the coating probably occurs after the ECL experiment. However, to assure reproducibility of the method, a new BTD-TPA film was prepared for every ECL experiment and never reused, as we have mentioned in the section 6 of the supplementary information (“BTD-TPA coating onto the electrode”, page S8): “A freshly prepared BTD-TPA-modified Pt wire was used for each ECL experiment.”

13. S23, Figure S11 (g) and (h). SEM images can be more convincing here.

We have replaced the photographs of the BTD-TPA coating onto the Pt electrode with 3D laser microscope images as reported in the new Supplementary Figure 14:

“**Supplementary Figure 14.** 3D laser microscope images of the Pt electrode covered with BTD-TPA film. (a) Pristine BTD-TPA film; BTD-TPA coating after the ECL experiment following the on/off pumping scheme – 10 sec pumping of the solution using the conditions of entry 3 in Table S8 and 10 sec stasis of the device – repeated for (b) 5 times and (c) 15 times.”

14. Page 15, line 398. Instead of “self-powered”, a better way to describe this system may be no electrical power or no external power supply, or hydro-powered or pressure driven. We thank the reviewer for the comment. We have revised the term “self-powered” in the conclusions section with the term “electricity-less”, since this lettering has been recalled several times in our manuscript.

15. This study only investigates TPrA, DBAE, and TEOA as co-reactants, what about other amines with clinical relevance, such as aromatic amines or trimethylamine? How much lower is the signal expected to be when these classes of amine are employed? We thank the reviewer for the comment. Following the reviewer’s suggestion, we have included in our study the ECL detection of several aromatic amines using the streaming

potential device (aniline, *N*-methylaniline and *N,N*-dimethylaniline). We have not included in the investigation trimethylamine because its electro-oxidation leads to the formation of a primary carbocation, that is a very reactive species of hazardous concern.

Our results have been included in the Supplementary Information document, as follows:

“19. Detection of aromatic amines

Considering the environmental relevance and human hazard of aromatic amines, the device was also tested for the ECL detection of this class of amines, including aniline, *N*-methylaniline and *N,N*-dimethylaniline. The results are summarized in Supplementary Table 12.

Supplementary Table 12. ECL intensity using different aromatic amines as coreactants (1 mM each) solubilized in an electrolyte-free MeCN/H₂O (3:1 v/v) solution at the flow rate of 0.30 mL/min. Phenolic resin monolith no. 2 was employed as the filling material. PMT bias = 1000 V.

Entry	Coreactant 1 mM	ΔP [MPa]	E_{str} [V]	i [μA]	Maximum ECL intensity [a. u.]	Integrated ECL intensity [a. u.]	pH [†]	Conductivity [mS/m]
1	Aniline	2.8	4.90	0.49	0	0	8.83	0.10
2	N -methylaniline	2.8	4.74	0.47	0	0	9.21	0.13
3	N,N -dimethylaniline	2.9	4.89	0.49	0	0	8.93	0.16

[†]Measured pH with a conductivity value less than 10 mS/m is to be considered as a reference value.”

In the manuscript, in the section “Detection of amines by such electricity-less ECL system”, a discussion of these results has also been included as follows:

“We also explored the use of our device for the ECL detection of several aromatic amines, since they are chemicals of concern for the environment, and can cause acute toxicity and bioaccumulation in humans and ecosystems.^{43,44,45} Aniline, *N*-methylaniline and *N,N*-dimethylaniline were investigated as coreactants for ECL generation in the streaming potential device by flowing an electrolyte-free MeCN/H₂O (3:1 v/v) solution containing 1 mM of each target aromatic amine at the flow rate of 0.30 mL/min. As summarized in

Supplementary Table 12, all aromatic amines generated similar ΔP , E_{str} and i values, indicating that the different aromatic structure did not influence the physical state and operation of the device. Interestingly, the conductivity values of these solutions were considerably lower compared to the values obtained using aliphatic amines as coreactants (comparison of conductivity data between Supplementary Tables 11 and 12), a fact that is, however, beneficial for the buildup of streaming potential in our system. Despite these positive findings, no ECL emission was detected for all the three investigated aromatic amines. This result is not surprising, since it is well known that aromatic amines do not show any ECL activity. Because the aromatic structure of the amine can delocalize the positive charge formed after electro-oxidation through resonance, the intermediate radical species tend to be excessively stabilized and, hence, less reactive towards the oxidized chromophore, hindering the ECL reaction.^{54,57} It should also be noted that aromatic amines, such as anilines, have been reported to behave as quenchers of the ECL emission,⁵⁸ likely due to an energy transfer mechanism from the emitting excited state to electro-oxidation products of the amine.⁵⁹”

Please, kindly refer to the manuscript for the listed references.

Reviewer #4 (Remarks to the Author):

We would like to acknowledge the early career researcher for the time and the efforts to co-review our manuscript.

Reviewer #5 (Remarks to the Author):

This study presents a novel micro-fluidic device utilizing bipolar ECL detection driven by streaming potential. The device is a simple and low-cost analytical tool that operates without external energy sources. While this research is innovative and appealing, the evaluation of the sensing system's anti-interference capabilities is limited. I recommend acceptance of the article following more comprehensive investigations into anti-interference performance using actual samples.

We thank the reviewer for the positive evaluation of our work and for the comment.

Following the reviewer's suggestion, we have performed a new series of experiments in order to evaluate the anti-interference performances of the device towards the sensing of amines.

We have reported the description of our method and the data obtained in the Supplementary Information document, as follows:

“22. Evaluation of the anti-interference capabilities of the device

The anti-interference performances of the streaming potential device were additionally investigated in both the standard electrolyte, composed of MeCN/H₂O (3:1 v/v) without supporting electrolyte, and tap water. To this purpose, the ECL detection of multiple amines solubilized in the same solution sample, or with the addition of other analytes, e.g. salt, was carried out using the device. The results are summarized in Supplementary Tables 15 and 16 and Supplementary Figs. 19 and 20.

Supplementary Table 15. Anti-interference performances study based on the ECL detection of multiple amines (coreactants) solubilized in the same solution sample composed of an electrolyte-free MeCN/H₂O (3:1 v/v) solution at the flow rate of 0.30 mL/min. Phenolic resin monolith no. 2 was employed as the filling material.

Entry	Multiple amines system (coreactants)	ΔP [MPa]	E_{str} [V]	i [μA]	Maximum ECL intensity [a. u.]	Integrated ECL intensity [a. u.]	pH [†]	Conductivity [mS/m]
1	1 mM TPrA	2.5	6.41	0.73	0.093	16.24	9.62	3.96
2	1 mM TPrA + 1 mM DBAE	2.5	6.29	0.73	0.030	4.28	9.87	3.62
3	1 mM TPrA + 1 mM TEOA	2.7	6.36	0.72	0.005	0.59	9.77	3.17
4	1 mM TPrA + 1 mM Aniline	2.7	6.17	0.73	0	0	9.59	3.06
5	0.5 mM TPrA + 0.5 mM DBAE	2.8	6.13	0.72	0.068	12.19	9.44	2.95
6	0.25 mM TPrA + 0.25 mM DBAE	3.1	6.48	0.72	0.290	48.46	9.34	1.92
7	2 mM TPrA	3.2	6.24	0.72	0.028	3.73	10.58	1.91

[†]Measured pH with a conductivity value less than 10 mS/m is to be considered as a reference value.

Supplementary Figure 19. ECL intensity of the BTD-TPA film in air during the pumping of the solution for 360 s with the streaming potential device using an electrolyte-free MeCN/H₂O (3:1 v/v) solution at the flow rate of 0.30 mL/min having a coreactant system composed of (a) the coexistence of TPrA with other aliphatic or aromatic amines and (b) TPrA and DBAE in different concentration ratios. Phenolic resin monolith no. 2 was employed as the filling material. PMT bias = 1000 V.

Supplementary Table 16. Anti-interference performances study based on the ECL detection of 1 mM TPrA solubilized in only tap water and in tap water containing 3% NaCl (0.5 M) at the flow rate of 1.0 mL/min. The addition of salt in tap water simulates water with high salinity similar to that of sea water. Phenolic resin monolith no. 2 was employed as the filling material.

Entry	Solution system (tap water)	ΔP [MPa]	E_{str} [V]	i [μA]	Maximum ECL intensity [a. u.]	Integrated ECL intensity [a. u.]	pH	Conductivity [mS/m]
1	1 mM TPrA	4.3	1.54	0.54	0.020	2.57	8.6	33
2	1 mM TPrA + 3% NaCl (0.5 M)	4.6	0	0	0	0	8.9	4.5×10^3

Supplementary Figure 20. ECL intensity of the BT-D-TPA film in air during the pumping of the solution for 360 s with the streaming potential device using tap water as the solvent containing only 1 mM TPrA (dark blue line) and 1 mM TPrA with the addition of 0.5 M NaCl (light orange line) to simulate water with high salinity. Flow rate was 1.0 mL/min. Phenolic resin monolith no. 2 was employed as the filling material. PMT bias = 1000 V.”

We have included a discussion of these results in the manuscript under the section “Electricity-less ECL generation in fully aqueous systems”, as follows:

“We lastly evaluated the anti-interference capabilities of the streaming potential device for sensing amines. We initially performed this study by flowing in the device the standard electrolyte, namely the electrolyte-free MeCN/H₂O (3:1 v/v) solution, containing a mixture of multiple amines, generally TPrA and another aliphatic or aromatic amine, at the flow rate of 0.30 mL/min. The ECL detection of such multiple coreactant systems was then compared with the ECL signal of the standard solution containing only 1 mM TPrA. When the flowing electrolyte contained in the same concentration TPrA and another aliphatic amine, such as DBAE or TEOA, the resulting ECL signals were considerably lower compared to the solution containing only TPrA, even though E_{str} , currents and conductivity values were similar among the cases (Entries 1-3 in Supplementary Table 15 and Supplementary Fig. 19a). If an aromatic amine, like aniline, coexisted in the solution with TPrA, no ECL emission was observed, owing to the quenching effect of the aromatic species towards the emitting excited state⁵⁹ (Entry 4 in Supplementary Table 15 and yellow line in Supplementary Fig. 19a). However, an enhancement of the ECL emission was observed by decreasing the concentration of both TPrA and DBAE in the flowing solution (comparison between Entries 2, 5 and 6 in Supplementary Table 15 and ECL curves in Supplementary Fig. 19b). Because a decrease of the amine concentrations gives rise to a decrease of the solution conductivities and generates larger pressure drops, this is likely advantageous for the retention of streaming potential in the device. However, even if low solution conductivity was also observed by increasing the concentration of TPrA from 1 to 2 mM, the drastic decrease of the ECL intensity (Entry 7 in Supplementary Table 15 and blue line in Supplementary Fig. 19a) highlights the complex balance of conditions suitable for the generation of both streaming potential and ECL.

We finally evaluated the anti-interference performances of the device using tap water with the addition of 1 mM TPrA as the flowing solution. We also studied the effect of other analytes in the solution, such as NaCl, that in 0.5 M concentration simulates the high salinity typical of sea water. Because of the relatively high ionic conductivity of these solutions, the device was operated at the maximum flow rate of 1.0 mL/min in order

to achieve satisfactory E_{str} values. As summarized in Supplementary Table 16 and Supplementary Fig. 20, ECL emission could be detected in tap water containing only TPrA, but the flow of this solution generated higher pressure drop compared to the electrolyte-free MeCN/H₂O (3:1 v/v) solution containing the same TPrA concentration (comparison of ΔP values in Entries 1 in Supplementary Tables 15 and 16). The addition of NaCl completely suppressed the generation of streaming potential, owing to the extremely high ionic conductivity of the solution according to Smoluchowski's equation (Fig. 2a), and ECL reaction could not be initiated in this case.”

REVIEWER COMMENTS

Reviewer #1:

The authors have addressed my comments adequately. All the very best to the authors!

We thank again the reviewer for his/her kind suggestions to improve our work.

Reviewer #5:

In the revised manuscript, the authors conducted additional research on the anti-interface performance of the device. Experimental results indicate that the electrolyte significantly affects detection performance, and the fabricated device is only suitable for use in low salt concentration systems. I recommend that the manuscript be accepted in its current revision.

We thank again the reviewer for his/her kind suggestions to improve our work.

Reviewer #6:

The manuscript describes a microfluidic electrochemiluminescence (ECL) platform utilizing streaming potential for amine detection, with an emphasis on energy-efficient operation and portability. While the concept of eliminating external electricity through streaming potential-driven bipolar electrochemistry is interesting, the level of insight and innovation is somewhat lacking, particularly in comparison to the standards expected for publication in Nature Communications. I do not think that introducing a microfluidic system, consisting of a microchannel and pump, to replace the electrochemical workstation, is necessary. The workstation in fact can be even discarded by using an external voltage such as a battery. Portable ECL devices powered by batteries are already a mature and well-established technology. In fact, replacing a battery with a pump introduces unnecessary complexity, potentially complicating a relatively straightforward issue.

We thank the reviewer for his/her critical evaluation of our work.

We are aware that there are other simpler or more convenient ways to achieve ECL emission; indeed, Hogan and co-workers achieved ECL detection by using one of the most ubiquitous tools of our modern epoch, such as a smartphone (please, see ref. 17 in the manuscript: *Sens. Actuators, B* **216**, 608-613 (2015), doi.org/10.1016/j.snb.2015.04.087).

By introducing this work to the scientific community, our aim is to report a novel approach to trigger the ECL reaction, that is by a **streaming potential** induced by a **pressure gradient** in a bipolar electrochemistry device, and to show a possible application of this system for analytical purposes. Our study, which is the first demonstration of this concept, simply represents a different case compared to the common bipolar ECL systems where the luminescent reaction is triggered by an **externally applied electric field**, and, therefore, it represents an advancement in the fields of bipolar electrochemistry, ECL, and in the analytical science in general. We are strongly convinced that it is worthy to be reported. We are not seeking “competition” or discussion about “pros and cons” among our methodology and others. On the other hand, a new methodology may address new challenges, while rising additional perspectives to address, but this is the aim of the original research, isn't it?

Additionally, the authors repeatedly emphasize that the device can operate without electricity. I wonder whether the pump used for driving the fluid requires electricity. If the pump consumes electricity, it is not entirely accurate to claim that this system is truly “electricity-free”. If it can operate manually, how can the reproducibility and accuracy of the measurements be ensured.

As reported in our manuscript, we have used a plunger pump to induce the pressure difference that, in turn, leads to the buildup of streaming potential and we have applied such methodology to achieve the ECL detection of amines in water samples. The plunger pump utilizes electricity for operation. Nevertheless, in the section “Generation of ECL powered by streaming potential” of our manuscript, we have shown that a simple syringe operated manually is capable to generate streaming potential values similar to the ones obtained by using the plunger pump and, hence, being potentially suitable for the detection. Therefore, in such a case, it is possible to claim that the system can be operated in an “electricity-free” condition. Nevertheless, we have not applied the syringe-type system for amines detection.

Considering the reviewer’s remark, we have replaced the term “electricity-less” in two subsections of our manuscript with the more general “streaming potential” term:

_ “Detection of amines by such electricity-less ECL system” has now become “Detection of amines by such **streaming potential** ECL **device**”

_ “Electricity-less ECL generation in fully aqueous systems” has now become “**Streaming potential** ECL generation in fully aqueous systems”.

The only “electricity-less” term of the manuscript appears in the part of the text where we discuss about the generation of streaming potential by using a syringe operated by hand.

In addition, we have added the following discussion in the Conclusion section of the manuscript:

“We believe that our prototype **may** represents an innovative class of low-cost **and** portable **analytical** electrochemical devices that can be employed **by using the electrical power of nature**. Indeed, our future envision is that, **once this technology has advanced and become more solid, a continuous natural water flow, e.g., a river flow, can be exploited to produce the necessary electrical energy to run the device**. This would realize an electroanalytical system with integrated ECL detection that could function **without anthropic-derived electricity, with potential utility** in emergency situations or in

areas with limited access to electricity.”

Furthermore, as depicted in the manuscript, the streaming potential is affected by multiple factors, including flow rate, conductivity and pH of liquid. When applied to real-world samples, these variables could significantly impact the analytical performance of the device, potentially leading to inconsistent or unreliable results. On the other hand, the detection of coreactant-type analytes via ECL has been extensively reported in prior work (e.g., *Analyst*, 2021, 146, 5198; *ChemElectroChem* 2020, 7, 1207; *Chem. Commun.*, 2016, 52,12845).

We partially agree with the reviewer’s viewpoint.

We are also aware that the current version of this technology may be difficult to apply for the analysis of real samples, as the streaming potential output is intimately related to some solution’s parameters, as the reviewer mentioned. Indeed, our system represents a prototype of this methodology and there is undoubtedly room for improvement. On the other hand, we have shown in our work that once the solution parameters are set correctly, including the concentration of the amine, the results obtained from this device are reproducible and reliable. Therefore, it is theoretically possible to collect several different data by setting up the device with the diverse parameters, so that it may be calibrated for the analysis of different type of samples. In addition, all the affecting parameters are precisely known before starting the analysis: the flow rate is controlled by setting up the plunger pump before operation, while the conductivity and pH of the solution are measured before injecting the solution in the device. In our study, we have exemplified the device operation towards the analysis of real samples by analyzing tap water.

Furthermore, the inherent selectivity limitations of ECL for amine detection (a well-recognized challenge in the field) are neither partially addressed nor mitigated in this work. Given these concerns, I do not recommend this manuscript for publication in *Nature Communications*.

As we stated several times in the manuscript and in this reply letter, our device is a prototype and it represents the first example of ECL detection driven by streaming

potential. Therefore, additional strategies to expand the device applicability and versatility, and to address limitations on its selectivity, are possible in the future. For example, this technology can be coupled with other analytical techniques to improve the selectivity towards the different amines. In addition, a different ECL emitter (chromophore) can be employed to design an ECL system in which the quenching of the light emission can be exploited for the detection of aromatic amines (another possible strategy to improve the selectivity of the device). In this study, we have exclusively employed BTD-TPA as the ECL system, but the device can also be designed with other ECL emitters.

General comments:

1. I disagree with the sentence “It should also be important to underline that part of the great success of ECL is also due to the simple and relatively inexpensive equipment necessary to perform an ECL assay”. Compared with photoluminescence and electrochemical techniques, ECL instruments are generally more complex. And the most successful applications of ECL, namely the automated ECL immunoassays, rely on sophisticated and advanced equipment rather than simple or inexpensive setups.

We thank the reviewer for his/her remark.

We respectfully disagree with the point of view expressed by the reviewer. Depending on the type of study or the application one wants to achieve, the ECL equipment can be relatively simple or made more elaborated. For the ECL detection, like in our case, a dark room, a light collector and a voltage supply are sufficient to achieve the proposed aim. A basic potentiostat suitable to induced ECL *via* chronoamperometry or cyclic voltammetry techniques is far less expensive than a potentiostat bearing the module for Electrochemical Impedance Spectroscopy (EIS). Compared to other luminescent techniques, such as fluorescence, there is no need to use an external excitation light source to induce the excited state of the luminophore in the ECL case, as this is done through the applied voltage. In addition, all the optics necessary to align the excitation light source towards the sample (monochromator, lens, etc.) is not necessary in ECL, making its setup relatively simple and inexpensive compared to photoluminescence and other

electrochemical techniques.

On the other hand, it is true that ECL imaging studies, for instance of cells or subcellular constituents such as those reported by Sojic and coworkers, use a more complex and expensive setup generally composed of a highly sensitive camera (like an Electron Multiplying Charge-Coupled Device (EM-CCD)) coupled with a direct or inverted microscope (please, see as refs in this regard: *J. Am. Chem. Soc.* **2017**, *139*, 16830–16837, doi.org/10.1021/jacs.7b09260 ; *J. Am. Chem. Soc.* **2018**, *140*, 14753–14760, doi.org/10.1021/jacs.8b08080 ; *Angew. Chem. Int. Ed.* **2023**, *62*, e202218574, doi.org/10.1002/anie.202218574 ; *Anal. Chem.* **2023**, *95*, 7372–7378, doi.org/10.1021/acs.analchem.3c00869).

It is also absolutely true that the commercial ECL systems to whom the reviewer refers, that are the cobas and Elecsys systems developed by the Hitachi-Roche partnership, are extremely costly and bear a considerable sophisticated technological know-how to be developed. However, this is because these systems are specifically designed to be fully automated, so that to reduce sampling routing carried out by the human operator at the minimum level, and in continuous operation with high-throughput performances, so that to satisfy the workload in large hospital analysis (for example, the cobas e801 analytical unit can perform up to 300 tests per hour and it is compatible with immunochemistry and clinical chemistry modules, please visit this website for further information: <https://diagnostics.roche.com/global/en/products/instruments/cobas-e-801-ins-2202.html#productSpecs>). However, the ECL core of these instruments is composed of a microfluidic electrochemical cell and a photomultiplier tube (PMT), a common experimental setup for ECL analysis, like in our case (please, see: K. Erler “Elecsys[®] immunoassay systems using electrochemiluminescence detection”, *Wiener Klinische Wochenschrift* 1998:110 Suppl 3:5-10, Figure 5.).

Another successful commercial application of the ECL detection represented by Meso Scale Discovery (MSD) uses a simpler and less expensive equipment compared to the Hitachi-Roche systems. In a MSD system, carbon electrodes constitute the bottom of multi-array and multi-spot microplates that allows for the easy binding of biological reagents bearing electrochemiluminescent labels (please, visit this website for further information: https://www.mesoscale.com/en/technical_resources/our_technology/ecl).

Therefore, considering the versatility of setups and costs for the most diverse ECL applications, we think it is safe to claim that “**part** of the great success of ECL is also due to the simple and relatively inexpensive equipment necessary to perform an ECL assay” as we stated in our manuscript.

2. As author mentioned in Introduction: “the precise spatial and temporal control over the light emission reaction guarantees high selectivity, meaning that the detection of the target analyte can even be achieved in complex matrix such as urine or blood”. However, the good spatiotemporal control of ECL is not directly related to high detection selectivity. While it aids in enhancing detection sensitivity and enables remote ECL imaging of large objects, such as single cells, detection selectivity arises from specific recognition events—such as the binding between antigen and antibody, the selective catalytic reaction between enzyme and substrate, or complementary base pairing between nucleic acid strands.

We thank the reviewer for the remark and we apologize for the uncorrected sentence. According to the reviewer’s comment, we have revised this statement in the manuscript as follows: “Moreover, the precise spatial and temporal control over the light emission reaction enables the remote imaging of biological entities and the mapping of the electrochemical reactivity at the electrode surface.^{8–10}” and changed refs 8 to 10 as follows:

(8) Valenti, G., Scarabino, S., Goudeau, B., Lesch, A., Jović, M., Villani, E., Sentic, M., Rapino, S., Arbault, S., Paolucci, F. & Sojic, N. Single cell electrochemiluminescence imaging: from the proof-of-concept to disposable device-based analysis. *J. Am. Chem. Soc.* **139**, 16830–16837 (2017).

(9) Knežević, S., Han, D., Liu, B., Jiang, D. & Sojic, N. Electrochemiluminescence microscopy. *Angew. Chem. Int. Ed.* **63**, e202407588 (2024).

(10) Yan, Y., Ding, L., Ding, J., Zhou, P. & Su, B. Recent advances in electrochemiluminescence visual biosensing and bioimaging. *ChemBioChem* **25**, e202400389 (2024).

3. I also disagree with the sentence, “However, metallic electrodes are usually preferred because of their stability and fast kinetics of electron transfer reactions”. The metallic electrodes are more susceptible to electrochemical oxidation compared with carbon-based or metal oxide electrodes, which often exhibit superior stability under oxidative conditions. That is why in commercial immunoanalyzer the platinum working electrode must be repeatedly renewed prior to ECL detection.

As the reviewer mentioned, the electrochemical passivation of metallic electrodes (like platinum and gold), due to the generation of an oxide layer which hinders the coreactant oxidation, is a common drawback of metallic electrodes and this is the reason why the surface state must be regenerated before every ECL measurements. However, this characteristic, which allows the continuous operation of the same electrode for thousands of ECL cycles, like in the Hitachi-Roche Elecsys systems, is one of the reason why metallic electrodes are used in commercial systems (together with other peculiarities as we mentioned in the manuscript). Indeed, gold disk electrodes were initially employed in the first commercially available ECL instrument (Origen I analyzer) introduced by IGEN International (*Anal. Chem.* **2000**, *72*, 3223–3232, doi.org/10.1021/ac000199y , Experimental section on page 3224), and nowadays Elecsys instruments still use platinum (K. Erler “Elecsys[®] immunoassay systems using electrochemiluminescence detection”, *Wiener Klinische Wochenschrift* 1998:110 Suppl 3:5-10, section “Technical realisation” on page 7.). Therefore, there is any uncorrected term in our claim.

We have specified this discussion in the revised version of the manuscript: “However, metallic electrodes are usually preferred **in commercial systems** because of their stability and fast kinetics of electron transfer reactions.^{44,46}”, including the two above-mentioned manuscripts as refs. 44 and 46.

4. For ECL detection of analytes, the requirements of electrolytes/buffer solution, sample pretreatments, and chemical modification of the analyte environment are not necessarily drawbacks, as they do not compromise the quantitative accuracy of the results. In the case of detecting amines in pure samples, such as tap water, sample pretreatment is also not required in conventional portable ECL devices.

We thank the reviewer for the comment.

Considering the reviewer's point of view, we have rephrased the target discussion on page 11 of our manuscript as follows: "Moreover, since the detection of the target species can be accomplished without modification of the sample's chemical environment and does not require any preconcentration or derivatization steps, the use of the device for sensing applications is also rather simple and practical."

We have also rephrased a similar sentence in the "Discussion" section as follows: "Remarkably, the operation of the device in a fully aqueous solution is also rather simple and practical, since the streaming potential-driven ECL system can operate reliably without any sample's pretreatment step."

Specific comments:

1. How do the authors keep the batch-to-batch reproducibility of luminophore coating on the electrode? Please include characterization data such as electron microscopy (e.g., SEM or TEM) to support the consistency and uniformity of the coating.

We thank the reviewer for the comment.

Considering our experimental methodology, before every experiment, we used to prepare a fresh BTD-TPA film on the Pt wire, and its uniformity was confirmed through optical imaging in both ambient light conditions and under UV light irradiation. Cyclic voltammetry experiments confirmed that our method generated BTD-TPA films with a highly reproducible redox behavior. Optimization of the BTD-TPA loading on the Pt wire was also conducted in the frame of our study (please, see section 9 of the Supplementary Information document).

We have reported such information in section 6 of the Supplementary Information document, as follows:

"6. BTD-TPA coating onto the electrode

Benzothiadiazole-triphenylamine (BTD-TPA) was prepared according to the literature² and used as ECL chromophore. A Pt wire ($\phi = 0.6$ mm) was coated with BTD-TPA by dipping the metallic wire into a chloroform solution of BTD-TPA one time, and then dried at the air. A freshly prepared BTD-TPA-modified Pt wire was used for each ECL

experiment. An Olympus OLS4100 3D laser microscope was employed for the observation and evaluation of the BTD-TPA coating on the Pt wire.³

The adopted procedure enabled the formation of a uniform coating of BTD-TPA on the Pt wire (Supplementary Fig. 4) and cyclic voltammetry investigations showed the high reproducibility of the method among different samples (Supplementary Fig. 5).

Supplementary Figure 4. Optical images of three different BTD-TPA coatings (samples (a), (b) and (c)) prepared on three different Pt wires under ambient lighting (upper images of each series) and under UV light (excitation wavelength 365 nm) (lower images of each series) at different enlargements. The UV images particularly shows the uniform distribution of each BTD-TPA coating on the Pt wires.

Supplementary Figure 5. Cyclic voltammetry of the three different BTD-TPA coatings prepared on the three different Pt wires shown in Supplementary Fig. 4: sample (a) red line, (b) green line and (c) light blue line. Solution conditions (in air): MeCN/H₂O (3:1 v/v) containing 0.1 M Bu₄NPF₆. Scan rate: 0.1 V s⁻¹.

2. In the manuscript, it is mentioned that the weakening of light intensity is due to the physical or chemical deterioration of the BTD-TPA. While images of physical detachment have been provided later in the text, further evidence is needed to substantiate the chemical factors contributing to this deterioration. Please include additional experimental data or analysis to support the chemical degradation hypothesis.

We thank the reviewer for the remark.

All the investigations we performed to evaluate a possible chemical degradation/modification of the BTD-TPA film (i.e., NMR analysis) were inconclusive, and it has not

been possible to obtain a clear conclusion. Therefore, in the current version of our manuscript, we have removed any mention to the possible chemical degradation of the BTD-TPA film after ECL reaction. We have only kept the information related to the physical deterioration of the film, as we have evidence of this from the 3D laser microscope images as reported in Supplementary Fig. 16.

3. In Supplementary Table 2, the authors described that the minimum voltage required to trigger ECL is 0.23 V. However, this may not be accurate, as the triggering voltage is likely in between 1.0 V and 2.3 V. Please revise the statement or provide additional data to validate this claim.

In our manuscript, in the section “Generation of ECL powered by streaming potential” on page 9, we have reported a value of the triggering voltage (E_{str}) of **2.3 V**, not **0.23 V** as the reviewer stated. This is an average value of the three measurements of E_{str} in Entry 2 of Supplementary Table 2 (2.32 V, repeated twice, and 2.28 V) by using those experimental conditions. Therefore, our claim is correct and in agreement with the reviewer’s expectation.

According to this discussion, we have slightly revised this sentence in the manuscript, as follows: “**In these experimental conditions**, the minimum E_{str} required to trigger the ECL reaction was found to be 2.3 V (Entry 2 in Supplementary Table 2), **a value that is considerably larger** to the threshold potential obtained by the voltage-sweep ECL measurements with the two-electrode system using the same electrolyte conditions (Supplementary Fig. 5).”

4. In Supplementary Table 2, the E_{str} value for entry 4 is significantly higher than that for entry 3, yet there is no noticeable difference in their light intensities. Please provide further explanation or analysis to clarify this discrepancy.

We thank the reviewer for the critical observation.

We have performed again the three set of measurements of Entry 4 in Supplementary Table 2 and we have re-calculated the average integrated ECL intensity by using the new

set of data. In the revised version, Entry 4 shows an average value of integrated ECL intensity of 1.522 a.u., which is a middle value between 1.416 a.u. of Entry 3 (lower flow rate and E_{str} value) and 1.757 a.u. of Entry 5 (higher flow rate and E_{str} value). Now, the values of integrated ECL intensity of the six entries in Supplementary Table 2 are in line with the values of flow rate, E_{str} , pressure and current as expected. The value of Entry 4 has been revised in Supplementary Table 2.

5. The reaction mechanism between BTD-TPA and TPrA should be discussed in more detail. As shown in Figures 3e and 5a, why does the ECL intensity initially increase and then decrease with the increase of TPrA concentration? For instance, in Table 6, at a flow rate of 0.3 mL/min, the E_{str} and i values for 5 mM TPrA are higher than those for 1 mM TPrA, yet the light intensity decreases to 1/6.

We thank the reviewer for the comment.

In our manuscript on page 11, we have reported that “oxidation of both BTD-TPA and TPrA should occur at the anode with appropriate **balance**.” With the term “balance” we refer to the suitable oxidation of both emitter and amine species, which depends on coreactant concentration (and potential). We have specified this in the revised version of the manuscript, as follows: “Considering the solid-state ECL mechanism of BTD-TPA,³⁴ the concentration of the coreactant is a key factor, because the oxidation of both BTD-TPA and TPrA should occur at the anode with appropriate balance. In fact, it is well known that aliphatic amines can quench the excited state of aromatic molecules,⁴¹ and, depending on coreactant concentration and applied potential, this may also occur in the frame of the ECL process.⁴²” with the following refs:

(41) Bertocchi, M. J., Bajpai, A., Moorthy, J. N. & Weiss, R. G. New insights into an old problem. Fluorescence quenching of sterically-graded pyrenes by tertiary aliphatic amines. *J. Phys. Chem. A* **121**, 458–470 (2017).

(42) Barbante, G. J., Kebede, N., Hindson, C. M., Doeven, E. H., Zammit, E. M., Hanson, G. R., Hogan, C. F. & Francis, P. S. Control of excitation and quenching in multi-colour electrogenerated chemiluminescence systems through choice of co-reactant. *Chem. Eur. J.* **20**, 14026–14031 (2014).

6. As shown in Figures 5a, when the concentration of TPrA exceeds 0.5 mM, the ECL intensity decreases, suggesting that both low and high concentrations of TPrA may produce similar ECL signals. In this case, how do authors propose achieving accurate quantification of TPrA in the sample solution?

We thank the reviewer for the comment.

As the reviewer noticed, we have this kind of “volcano-type” calibration curve for TPrA detection. However, these data are obtained by using an unique flow rate. Therefore, by changing the flow rate, different calibration curves can be obtained, so that it is possible to achieve wider ranges of detection.

7. In Supplementary Figure 5, panel (c) exceeds the maximum range. Please redraw the figure to ensure that the data falls within the appropriate range.

We thank the reviewer for the remark.

We have performed again the CV-ECL measurement using the suitable range for the ECL intensity and we have revised it in Supplementary Figure 5, as follows:

8. For Ag and Cu, the E_{str} and i values are generally consistent with those of Pt and Au; however, no detectable light intensity is observed. I wonder whether the luminophore does not firmly adhere to the electrode. Or other reasons?

In our experiments, the BTD-TPA film could be easily formed on the Ag and Cu electrodes, in a similar manner to that of Pt and Au, and no film dissolution was observed during the operation of the device. Our conclusion is that, Ag and Cu oxidation, that generally occur at low oxidation potentials, are the predominant redox processes at the split BPE anode, causing a lack of the chromophore's oxidation.

We have reported this evidence in the manuscript, with slight revision to include this discussion, as follows: "ECL emission could not be detected using the less noble metals Ag and Cu, although the BTD-TPA films were firmly adhered on the electrodes and no apparent metal dissolution was observed. Nevertheless, the use of Cu generated the highest current flowing through the split BPE system (Entry 4 in Supplementary Table 7), an evidence that might be ascribed to copper oxidation that generally occurs at relatively mild potentials. Similarly, Ag oxidation, that also occurs at low anodic potentials, may be the predominant redox process at the split BPE anode, a fact that might prevent oxidation of the BTD-TPA chromophore leading to a lack of light emission. However, the different nature of the electrodes did not significantly impact the magnitude of E_{str} (Supplementary Table 7), since 0.2 V is the difference between the highest and the lowest E_{str} values generated from the use of Cu and Pt, respectively."

9. What is the concentration range of TPrA in aquatic ecosystems?

We have found the following literature reporting TPrA concentration ranges, or limit of detections, in water samples by using different detection methods:

- a) $5 \times 10^{-5} - 1 \times 10^{-2} \text{ mol l}^{-1}$ by flow injection analysis with chemiluminescence detection (see "Flow injection procedure for the determination of tertiary amines in water and sea water using chemiluminescence detection" *Analyst* **1989**, *114*, 1659-1661, doi.org/10.1039/AN9891401659)
- b) **0.26 $\mu\text{g/kg}$** in bottom sediments sampled in Tokyo Bay by using gas chromatographic-mass spectrometric determination (see "Gas chromatographic-mass spectrometric

determination of lower aliphatic tertiary amines in environmental samples” *J. Chromatogr.* **1993**, *642*, 395-400, doi.org/10.1016/0021-9673(93)80104-G)

- c) **0.014 mM** lowest detected concentration by capillary electrophoresis with indirect UV detection (see “Preconcentration of aliphatic amines from water determined by capillary electrophoresis with indirect UV detection” *J. Liq. Chrom. & Rel. Technol.* **1997**, *20(1)*, 79-100, doi.org/10.1080/10826079708010638)
- d) **0.4 nM** lowest detection limit by electrochemiluminescence detection (see “Porous graphene containing immobilized Ru(II) tris-bipyridyl for use in electrochemiluminescence sensing of tripropylamine” *Microchim Acta* **2016**, *183*, 1211-1217, doi.org/10.1007/s00604-016-1756-0)

10. How did the authors ensure the selectivity of the device for TPrA detection?

Our device is not a selective device for TPrA detection, but it can detect also other amines. We have shown in the anti-interferences study that the resulting ECL intensity depends on the type of amine present in the flowing solution. In addition, as stated in a previous response, it is possible to obtain a calibration curve for every amine studied.

11. Error bar is missing in Figure 2b, 2c and 5a.

We thank the reviewer for the comment.

Since there is a clear linearity of the data in both Figs. 2(b) and 2(c), we have proved this relationship by calculating the standard deviation about the regression (S_r) for both materials, and we have reported these values in the revised version of Fig. 2, as follows:

Fig. 2. Generation of streaming potential within the microfluidic device. (a) Illustration of the streaming potential measurement system using a voltmeter between the two electrodes and the Smoluchowski's equation. Pressure drop (ΔP) and streaming potential (E_{str}) were measured using the cotton wool filling or the phenolic resin monolith no. 1 filling during pumping of the MeCN/H₂O (3:1 v/v) solution containing 0.5 mM Bu₄NPF₆ and 5 mM TPrA. The relationship between (b) ΔP and flow rate and (c) E_{str} and ΔP . The black squares indicate the values measured with the phenolic resin monolith no. 1 as a filler, whereas the blue triangles represent the values measured with the cotton wool as a filler. Black and blue dotted lines represent the linear fitting. S_r indicates the standard deviation about the regression.

Regarding the missing error bars in Fig. 5(a), we have performed again this set of experiments (with the same experimental conditions), and revised them in both manuscript (Fig. 5(a)) and Supplementary Information (Supplementary Table 10). We have also revised the LOD (0.01 mM on page 14 of the manuscript) obtained with the new set of data and by using the integrated ECL intensity values for calculation and not the ECL maximum as we did in the previous version:

Fig. 5. ECL detection of amines in aqueous solutions using the streaming potential device. (a) Dependence of the ECL intensity on the concentration of TPrA using an electrolyte-free MeCN/H₂O (3:1 v/v) solution at the flow rate of 0.30 mL min⁻¹. (b) ECL emission profiles obtained during the flow of a MeCN/H₂O (3:1 v/v) solution containing different amines (1 mM each) at the flow rate of 0.30 mL min⁻¹. (c) ECL emission profiles obtained using a fully aqueous system as the electrolyte: distilled water or tap water solutions containing 1 mM TPrA were injected into the flow cell at the flow rate of 0.30 mL min⁻¹. All measurements were performed using the resin monolith no. 1 as the filling material.

Supplementary Table 10. Dependence of ECL intensity on TPrA concentration (range 1.0–0 mM) in an electrolyte-free MeCN/H₂O (3:1 v/v) solution. Flow rate was 0.30 mL min⁻¹ in all cases.

Entry	TPrA concentration [mM]	ΔP [MPa]	E_{str} [V]	i [μA]	Maximum ECL intensity [a. u.]	Integrated ECL intensity [a. u.]
1	1.0	4.2	11.75	1.41	0.1393	45.135
2	0.75	4.4	11.95	1.38	0.3616	70.126
3	0.50	4.2	11.65	1.3	0.4960	91.264
4	0.25	4.2	11.69	1.35	0.3052	61.570
5	0.10	4.3	11.27	1.27	0.0280	3.228
6	0.050	4.1	10.88	1.17	0.0049	0.637
7	0.010	4.5	9.34	1.01	0.0085	0.147
8	0	4.5	5.89	0.61	0	0

REVIEWERS' COMMENTS

Reviewer #6 (Remarks to the Author):

The authors have seriously addressed my comments and revised the manuscript. I can see clearly the authors' positive attitude and effort to improve the manuscript. However, based on my personal expertise in the field, my own judgement on the novelty and scientific contribution of the work, as well as the potential impact of the work, I insist that the manuscript does not meet the publication criteria of the journal. Nevertheless, I respect other reviewers' opinion and the decision of the editorial office, if they think the manuscript deserves publication.

We thank the reviewer for the time spent to assess the quality of our work, and for his/her comments and remarks, which allowed further improvement of our manuscript. We honestly respect his/her judgement regarding the suitability for publication. From our side, we can firmly declare that we have done our best at every stage of the manuscript process, with further considerable efforts during the several revision processes. We are, therefore, confident of the experimental results and solidity of our work, and we accept the final decision with clear minds.

Review of the article ‘An Electrochemiluminescence Device Powered by Streaming Potential for the Detection of Amines in Flowing Solution’: Manuscript No. - NCOMMS-24-32987-T

The following major remarks require explanation before the decision is made:

- Abstract, line: “...its potential application as a simple and low-cost analytical device that can be operated **without the need of any type of energy**. The device showed high reproducibility and satisfactory detection limits...”

In this case, electrical energy was needed to pump the flow (e.g., by a Syringe pump). I am curious to know by how the flow is produced in the absence of an energy flow that would produce streaming potential. The claims of "low cost" and "no need of any energy" are what make the entire work noble. The liquid itself needed some energy to flow.

- The detection of amine in the water is the foundation for the entire analysis. Nevertheless, the literature section fails to clarify why it came to be identified. I propose that the harmful effects of amines in water on the environment and human health be included. Please read the following literature: Treatment of hazardous organic amine wastewater and simultaneous electricity generation using photocatalytic fuel cell based on TiO₂/WO₃ photoanode and Cu nanowires cathode, Chemosphere, Volume 289, February 2022, 133119.
- In my opinion, paper-based detection is quite inexpensive and easy to use. Kindly review the following piece of work: Electrochemiluminescence Detection in Paper-Based and Other Inexpensive Microfluidic Devices, ChemElectroChem 2017, 4, 1594., <https://doi.org/10.1002/celec.201700426>

How does the author justify, in this context, that the current flow energy-requiring detection device is "simple," "low cost," and "required no-energy! (see abstract)"? Anyone can claim that the existing paper-based devices is easy to use, inexpensive, and capable of self-capillary driven flow.

Also, in the paradigm paper-based detection, optimization of lateral flow assay is important to increase the specificity and efficacy. The authors may look into the following article recently published in this context and can fortify the literatures cited in the introduction part.

Solute imbibition in paper strip: Pore-scale insights into the concentration-dependent permeability, Physics of Fluids 35 (12), 2023., <https://doi.org/10.1063/5.0177100>

- As of right now, more technical details regarding the detection don't seem particularly noteworthy because this technique has already been established in the literature: A Wireless Electrochemiluminescence Detector Applied to Direct and Indirect Detection for Electrophoresis on a Microfabricated Glass Device, Anal. Chem. 2001, 73, 14, 3282–3288; Chem. Rev. 2004, 104, 6, 3003–3036: March 20, 2004, <https://doi.org/10.1021/cr020373d>